# Fengbo: a Clifford Neural Operator pipeline for 3D PDEs in Computational Fluid Dynamics

**Alberto Pepe**[1,2,*], **Joan Lasenby**[1]
[1] Signal Processing and Communications Lab
Department of Engineering
University of Cambridge
Trumpington Street, Cambridge, CB2 1PZ, UK
`{ap2219,jl221}@cam.ac.uk`

**Mattia Montanari**[3]
[3] Impact Engineering Laboratory
Department of Engineering Science
University of Oxford
Parks Road, Oxford, OX1 3PJ, UK
`exet3790@ox.ac.uk`

[2] Intelligent Cloud Technologies Laboratory
Huawei Research Center Germany & Austria
Riesstraße 25, 80992 München, Germany

## Abstract

We introduce Fengbo, a pipeline entirely in Clifford Algebra to solve 3D partial differential equations (PDEs) specifically for computational fluid dynamics (CFD). Fengbo is an architecture composed of only 3D convolutional and Fourier Neural Operator (FNO) layers, all working in 3D Clifford Algebra. It models the PDE solution problem as an interpretable mapping from the geometry to the physics of the problem. Despite having just few layers, Fengbo achieves competitive accuracy, superior to 5 out of 6 proposed models reported in Li et al. (2024) for the *ShapeNet Car* dataset, and it does so with only 42 million trainable parameters, at a reduced computational complexity compared to graph-based methods, and estimating jointly pressure *and* velocity fields. In addition, the output of each layer in Fengbo can be clearly visualised as objects and physical quantities in 3D space, making it a whitebox model. By leveraging Clifford Algebra and establishing a direct mapping from the geometry to the physics of the PDEs, Fengbo provides an efficient, geometry- and physics-aware approach to solving complex PDEs.

## 1 Introduction

Many natural phenomena and complex systems, including electromagnetism and seismic waves, are governed by partial differential equations (PDEs). Solving these PDEs enables the prediction of a system's state evolution over time, which is valuable in applications such as stock price estimation and weather forecasting. While PDEs often provide precise models of these systems, they are typically too complex to solve analytically. Numerical methods, such as finite element analysis (FEA) and finite difference methods (FDM), are some of the well-established techniques for approximating solutions in complex geometries and handling boundary conditions Perrone & Kao (1975); Liszka & Orkisz (1980); Friswell & Mottershead (1995). However, these methods require significant computational resources, particularly when high-resolution solutions are needed.

In the past decade, machine learning (ML) methods have been applied to solve PDEs Carleo et al. (2019); Willard et al. (2020); Karniadakis et al. (2021). ML-based methods can be several orders of magnitude faster than traditional numerical approaches, enabling rapid simulations while maintaining acceptable accuracy. This is particularly useful for applications requiring real-time predictions, such as weather forecasting and fluid dynamics simulations. Most ML approaches blend physical laws with large datasets to efficiently approximate PDE solutions, significantly reducing computational costs. Neural operators, in particular, have gained attention as a novel approach to solving PDEs Li et al. (2020b;a); Kovachki et al. (2023). They extend the idea of neural networks by learning mappings between function spaces, rather than finite-dimensional vectors. Unlike conventional

---

*This work was initiated during an internship at [2] and was completed thereafter.

neural networks, which approximate specific solutions, Neural Operators aim to approximate the underlying operator of a PDE, allowing for generalization across different input conditions and configurations, making them highly efficient at solving PDEs.

**Fengbo**. This paper introduces a Neural Operator pipeline for computational fluid dynamics (CFD) cast entirely in Clifford Algebra. Named after the Taoist deity of the wind, Fengbo leverages the embedding of data within an algebra of choice in the form of *multivectors*, which are the fundamental objects in Clifford Algebra, to integrate physics and geometry data throughout the architecture. Its operators, layers and neurons are all expressed as multivectors in Clifford Algebra.

Multivectors are a linear combination of objects, e.g. points, vectors, and planes, which can be employed to represent geometrical shapes but also physical quantities (e.g. pressure and velocity fields). This representation allows for an expressive and flexible encoding of complex relationships, ultimately leading to a strong inductive bias to the neural network. This bias preserves geometric relationships between different quantities, can ensure equivariance under transformations Ruhe et al. (2024); Pepe et al. (2024a), and allows for more descriptive models Brandstetter et al. (2022); Pepe et al. (2024c). As a result, we can achieve high performance using far fewer parameters compared to conventional models.

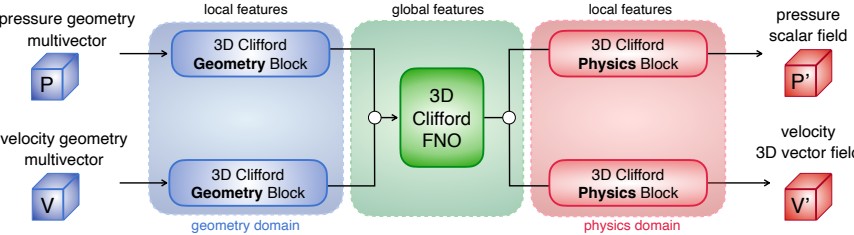

Figure 1: **The Fengbo architecture**. Irregular geometries are discretised into fixed-resolution volumes of multivectors, over which Fengbo operates. It consists of three steps: (i) The geometry blocks operate on the geometry of the PDEs domain, capture local features, ensure grade mixing and upsample the inputs; (ii) The Clifford FNO establishes a mapping between the PDEs geometry and their solution; (iii) The physics blocks operate on physical quantities, i.e. target of the regression. The entire architecture sits in 3D Clifford Algebra, guaranteeing interpretability.

Fengbo has three main components: (i) 3D Clifford Geometry block(s): one for each input geometry in the dataset, to mix elements of different grades in multivectors with geometrical meaning. (ii) 3D Clifford Fourier Neural Operator (FNO): to capture global interactions and map multivectors from the geometry to the physics domain. We extend their implementation in Brandstetter et al. (2022) to process full-grade 3D multivectors as opposed to only vector and bivector components. (iii) 3D Clifford Physics block(s): Similar to (i), but for multivectors with physical meaning. There is one Physics block for each output physical quantity to estimate in the dataset.

We tested Fengbo on the two available 3D computational fluid dynamics (CFD) datasets generated and analysed in Li et al. (2024). Fengbo takes input multivectors representing the shape of the vehicles and estimates the pressure field on their surfaces as well as the velocity field defined over the domain. It does so with fewer than half the parameters required by the GINO architecture (Li et al. (2024)) and by directly processing the geometries employed in CFD. Since every intermediate output in Fengbo is a multivector with geometrical or physical meaning, Fengbo is a whitebox model that allows for a clearer understanding of how data are processed and transformed from geometry to physics.

## 2 RELATED WORK

**Learning methods in PDE modelling**. A key challenge in applying machine learning (ML) to PDEs is ensuring that the model does not simply perform pattern recognition. Instead, the objective is for the model to capture the underlying physical principles governing the PDEs and accurately represent the geometry of the domain in which these equations are defined. Consequently, most models designed for PDEs are designed to address these requirements.

Physics-Informed Neural Networks (PINNs) Raissi et al. (2017; 2019); Cuomo et al. (2022), for example, do so by incorporate the governing PDEs into the neural network training process and learn directly from them. This integration helps ensure that the solutions respect physical constraints and produce realistic outcomes, addressing the limitation of simpler ML models that might fail to generalise on unseen data. However, PINNs are limited to a specific PDE and often require an additional Graph Network for spatial generalisation.

A similar philosophy is found in solver-in-the-loop methods Um et al. (2020); Brahmachary & Thuerey (2024); Lippe et al. (2024). These hybrid methods combine an ML architecture with a relatively simple numerical solver. The numerical solver helps to refine the predictions of the ML model, ensuring that the solutions remain grounded in the physical constraints of the problem. Deep Galerkin Method (DGM) algorithms Sirignano & Spiliopoulos (2018); Li et al. (2022a); Al-Aradi et al. (2022) also fall in the same category. DGM algorithms are trained to satisfy the differential operator, initial conditions, and boundary conditions, proving to be particularly suitable to deal with high-dimensional PDEs.

As CFD structures are often represented via point clouds, several approach exist in the literature based on extensions of the PointNet Qi et al. (2017a); Kashefi et al. (2021); Nemati Taher & Subaşı (2024); Kashefi (2024) and PointNet++ architectures Qi et al. (2017b); Zhang & Cao (2024); Gao et al. (2024). PointNet is a simple yet effective way to handle irregular point clouds without resorting to grids, but it fails at capturing global geometric context. PointNet++ mitigates this issue via multi-scale data processing at the expense of a higher computational cost.

Albeit versatile and flexible, PointNet-based methods are less accurate compared to more advanced PDE surrogates, including Transformer-based models Cao (2021); Li et al. (2022b); Xiao et al. (2023); Wu et al. (2024). Attention layers, especially when tailored to PDEs, offer a significant accuracy boost while keeping the model size small. The major drawback of Transformers is their computational complexity, generally $\mathcal{O}(N^2)$. Several attempts have been made in order to reduce the computational cost of such models, most notably the Galerkin Transformer Cao (2021), which reduces the cost of the quadratic Fourier-attention from $\mathcal{O}(N^2d)$ down to $\mathcal{O}(Nd^2)$, with $d$ the dimensionality of feature space, and the Transolver Wu et al. (2024), that introduces the Physics-Attention layer to learn over slices of the domain $\Omega_D$, with complexity of $\mathcal{O}(f(N;\theta))$, with $f$ being a function linear in $N$ with dependence on the model parameters $\theta$. Given the significant computational complexity of Transformer models, we regard them as a distinct category and instead focus our analysis on Neural Operators, of which Fengbo is an example. To justify our choice, a discussion on computational complexity of such models and their comparison to Fengbo is provided in Appendix E.

Neural operators have recently emerged as a key architecture to tackle the problem of PDE modelling Li et al. (2020a;b); Lütjens et al. (2022); Raonic et al. (2024); Azizzadenesheli et al. (2024). Neural operators differ from neural networks since they learn mappings between function spaces, or domains, instead of being function approximators like neural networks. When tackling PDEs, Neural Operators learn a mapping from input functions, which represent the initial or boundary conditions, to output functions, which represent the solution to the PDEs. They come come in several versions: Fourier Neural Operators, for example, operate in the frequency domain, where convolutions are more efficient at capturing long-range dependencies and periodic patterns in the data Li et al. (2020a; 2023; 2024), but Convolutional, Laplace and Graph Neural Operators have also been reported in the literature to address specific problem requirements. GINO Li et al. (2024), for example, is a pipeline combining a Graph Neural Operator, that handles irregular shapes and maps them onto a regular grid in latent space, and a Fourier Neural Operator, that processes the transformed input in latent space, that achieves state-of-the-art performance on large scale 3D PDEs.

Hybrid methods that combine Neural Operators and Transformers also exist, such as a the general Neural Operator Transformer in Hao et al. (2023), which introduces the heterogeneous normalized attention and the geometric gating mechanism for 2D PDEs. Such methods, however, are complex and up to $\times 4$ times larger than most Transformer-based models as shown in Wu et al. (2024).

**Clifford Algebra Networks**. Clifford Algebra introduces *multivectors* to extend linear algebra into a framework designed to couple multidimensional data and geometric transformations. Clifford Algebra has been shown to be a valuable resource in several fields, including physics, computer vision and computer graphics Lasenby & Lasenby (2001); Lasenby & Doran (2001); Doran & Lasenby

(2003); Dorst & Lasenby (2011). We present a brief overview of Clifford Algebra we use in this paper in Appendix A. Clifford Algebra Networks are architectures that work with multivector-valued inputs, outputs, weights and biases, and that can perform geometric transformations in Clifford Algebra. The renewed interest in this type of networks arose precisely due to their potential in PDE modelling, but they have demonstrated promising results in several other fields, including computer vision and bioinformatics Brandstetter et al. (2022); Roy et al. (2024); Ruhe et al. (2024); Pepe et al. (2024d;a;c;b); Hockey et al. (2024). By encoding the geometry and physical properties directly into the algebra, they can represent and solve PDEs by capturing the relationships between variables in a geometrically meaningful way. This approach allows for smaller yet more expressive and descriptive models that can better generalise the PDEs solution.

## 3 METHOD

**Notation.** Unless stated otherwise, we will employ lowercase Latin letters for scalar quantities (e.g. $p_1, v_1$), boldface Latin letters for vectors (e.g. $\mathbf{x}, \mathbf{n}, \mathbf{p}, \mathbf{v}$), uppercase Latin letters for multivectors (e.g. $P, V, Q, B, W$) or integers (e.g. $K, N, M, C$), lowercase Greek letters for real-valued maps (e.g. $\phi, \psi$) and uppercase, boldface Greek letters for multivector-valued maps (e.g. $\boldsymbol{\Phi}, \boldsymbol{\Xi}$). We use a dash symbol to distinguish multivectors describing geometrical quantities from those describing physical ones (e.g. $P, P'$).

**Navier-Stokes equations.** The Navier-Stokes equations describe the motion of fluids. They read as follows:

$$\frac{\partial \rho}{\partial t} + \nabla \cdot (\rho \psi) = 0 \tag{1}$$

Eq. 1 represents the conservation of mass in a fluid flow. It states that the time derivative of the fluid density $\rho$ plus the divergence of the mass flux $\rho\psi$ must be zero. Here $\psi$ represent the fluid velocity. This ensures that mass is conserved within the fluid domain.

$$\rho \frac{\partial \psi}{\partial t} + \rho(\psi \cdot \nabla)\psi = -\nabla\phi + \mu\nabla^2\psi + \mathbf{f} \tag{2}$$

Eq. 2 describes the conservation of momentum. It accounts for: the time rate of change of momentum $\rho\frac{\partial \psi}{\partial t}$, the convective term $\rho(\psi \cdot \nabla)\psi$ which represents the transport of momentum due to the fluid's velocity, the pressure gradient force $-\nabla\phi$, the viscous forces $\mu\nabla^2\psi$ which resist the flow of the fluid, and external forces $\mathbf{f}$ applied to the fluid. Here $\mu$ is the dynamic viscosity of the fluid and $\phi$ is the pressure. As in many CFD applications, Eq. 2 can be simplified by assuming that the fluid is incompressible and no forcing terms are present. We shall restrict this work to the steady state model, in which partial derivatives in time are null.

### 3.1 ARCHITECTURE

The Fengbo architecture, shown in Fig. 1, is an architecture that maps the geometry of the domain of the PDEs onto their solution. Specifically, we are interested in estimating jointly the scalar pressure field $\phi(\mathbf{x}) : \Omega_D \subset \mathbb{R}^3 \to \mathbb{R}$ and in the vector velocity field $\psi(\mathbf{x}) : \Omega_D \subset \mathbb{R}^3 \to \mathbb{R}^3$ that satisfy Eq. 1-2 over an irregular domain $\Omega_D$. We do as follows:

**Voxelisation of the fluid domain.** To deal with an irregular domain we need to support unstructured meshes which are commonly used in CFD. We do so by generating a regular grid of $M \times M \times M$ voxels inside the fluid domain $\mathcal{D}$. In general the voxels do not fit the boundary and the discretization parameter $M$ should be sufficiently large to capture a good level of geometric details. Our domain is a discrete volume of 3D space throughout, a simpler alternative to embedding in latent space or the use of a graph representation of data.

**Clifford Algebra embedding.** We fill in each voxel $i, j, k \in \mathcal{D}$ with a multivector $P : \mathcal{D} \to G(3, 0, 0)$ (see Appendix A for notation), in which $\mathcal{D} \subset \mathbb{R}^3$ represents the discrete grid of voxels in which the multivector field is defined and $G(3, 0, 0)$ is the 3D Clifford Algebra. We call $P_{ijk}$ the multivector associated with the voxel specified by indexes $i, j, k$. We construct multivectors $P$ to have a *scalar* component $m_p$, a *vector* component $\mathbf{p}$ and a *bivector* component $B$.

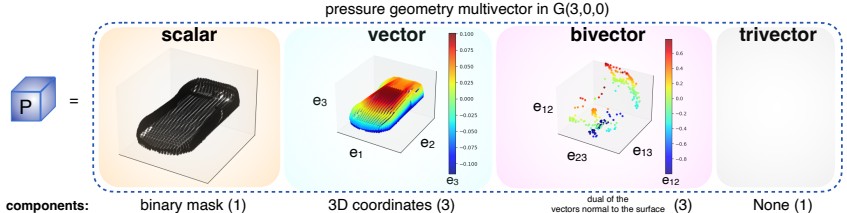

Figure 2: An example of pressure geometry multivector $P$. It has a scalar component (the binary mask $m_p$), 3 vector components (the 3D coordinates $\mathbf{p}$) and 3 bivector components (the dual of the vectors $\mathbf{n}$ normal to points $\mathbf{p}$).

- The scalar part is a binary mask $m_p$, included to inform the network about which voxels are filled and avoid ambiguity between the origin and empty voxels in the vector part, since for both we have that $p_1 = p_2 = p_3 = 0$.

- The vector part $\mathbf{p}$ represents the $N$-point point cloud $\mathbf{p} \in \mathbb{R}^3$ of coordinates in 3D space and it encodes information about the *shape* or contour of the object.

- The bivector $B$ represents the plane orthogonal to the normal $\mathbf{n}$ defined for each point in $\mathbf{p}$. In other words, $B$ is the dual of $\mathbf{n}$, i.e. $B = I_3\mathbf{n}$, in which $\mathbf{n}$ is the normal vector perpendicular to the mesh points on the car surface and $I_3 = e_1 \wedge e_2 \wedge e_3$ is the $G(3, 0, 0)$ pseudoscalar. $B$ is an oriented plane and hence it can be interpreted as containing information about the *surface* of the object.

- The trivector component is left blank.

Since the output pressure field $\phi(\mathbf{x})$ is defined at each point $\mathbf{p}$, we call $P$ the **pressure geometry multivector**. An example of a pressure geometry multivector is given in Fig. 2. The general form of the pressure geometry multivector is:

$$P = m_p + \mathbf{p} + B = \underbrace{m_p}_{\text{scalar}} + \underbrace{p_1e_1 + p_2e_2 + p_3e_3}_{\text{vector}} + \underbrace{B_{12}e_{12} + B_{13}e_{13} + B_{23}e_{23}}_{\text{bivector}} \qquad (3)$$

For datasets that include other physical fields, we define other input multivectors. For example, if the velocity vector field $v$ is known, we construct a corresponding multivector $V$. The multivector $V : \mathcal{D} \to G(3, 0, 0)$ is also defined over a regular grid of voxels. We construct $V$ to have a *scalar* component $m_v$ and a *vector* component $\mathbf{v}$. The vector component $\mathbf{v}$ corresponds to the $K$-point point clouds $v \in \mathbb{R}^3$, with $K \gg N$, defined for points *surrounding* the car surface, and $m_v$ is its corresponding binary mask defined similarly to $m_p$. Since the output velocity field $\psi(\mathbf{x})$ is defined over each point of $\mathbf{v}$, we call $V$ the **velocity geometry multivector**. Each velocity geometry multivector $V$ is of the form:

$$V = m_v + \mathbf{v} = \underbrace{m_v}_{\text{scalar}} + \underbrace{v_1e_1 + v_2e_2 + v_3e_3}_{\text{vector}} \qquad (4)$$

in which, similarly to $P$,

- The scalar part is a binary mask $m_v$.

- The vector part $\mathbf{v}$ represents the $N$-point point cloud $\mathbf{v} \in \mathbb{R}^3$ of coordinates in 3D space over which the velocity field is defined.

- The bivector and trivector components are left blank.

An example of $V$ is shown in Fig. 3. The geometry multivectors $P$ and $V$ are representative of the geometry simply because they *are* themselves the geometry of the PDEs domain.

**3D Clifford Geometry block.** We define the 3D Clifford Geometry block to be the module acting on volumes of multivectors with a sequence of three 3D convolutional layers in Clifford Algebra.

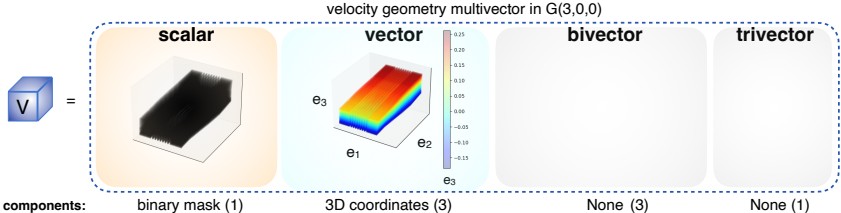

Figure 3: An example of velocity geometry multivector $V$. It has a scalar component (the binary mask $m_v$) and 3 vector components (the 3D coordinates $\mathbf{v}$).

In Clifford Algebra Networks, weights and biases are multivectors $W, D \in G(3, 0, 0)$, and convolutions are performed via geometric products:

$$Q_{ijk}^{(c_{out})} = \sum_{c_{in}=1}^{C} \sum_{l} \sum_{m} \sum_{n} P_{i+l,j+m,k+n}^{(c_{in})} W_{ijk}^{(c_{in},c_{out})} + D_{ijk}^{(c_{out})} \tag{5}$$

where the range of the summation of $l, m, n$ is specified by the kernel size and $c_{in}, c_{out}$ are the input and output channels, respectively. The 3D Clifford Geometry block takes in input a single geometry multivector (e.g. $P, V$) and it outputs $C$ channels of multivectors. It serves four purposes:

- grade mixing: multivectors $P, V$ in input to it only contains elements of a certain grade. Convolutional layers allow grades to mix and have full multivectors in 3D Clifford Algebra (i.e. with scalars up to trivector components).
- capturing local interactions: convolutions are traditionally used to extract feature from data which are close to each other in space.
- filling up the volume: fitting an irregular grid into a regular one requires a high-resolution grid, meaning that most of the initial input volume is sparse. Convolutions with a large enough kernel avoid this issue by filling up the volume.
- increasing the number of channels in input to the Clifford FNO block.

We refer to the output of the geometry block processing the shape over which the pressure field is defined (i.e. the pressure multivector $P$) as $Q_P$ and to the output of the geometry block processing the shape over which the velocity field is defined (i.e. the velocity multivector $V$) as $Q_V$.

**Clifford FNO block.** The 3D Fourier Neural Operator (FNO) in Clifford Algebra learns a multivector valued function $\mathbf{\Phi}(Q) : G(3, 0, 0) \rightarrow G(3, 0, 0)$ described by the 3D Clifford convolution theorem of Brandstetter et al. (2022):

$$Q' = \mathbf{\Phi}(Q) = \mathcal{F}^{-1}\{\mathcal{F}\{Q\}(\xi) \cdot \mathcal{F}\{\mathbf{k_a}\}(-\xi)\}, \tag{6}$$

in which $\mathbf{k_a} : \mathbb{R}^3 \rightarrow G(3, 0, 0)$ is the learnable filter of the FNO and $\mathcal{F}$ and $\mathcal{F}^{-1}$ are the Fourier and inverse Fourier transforms, respectively, with the Fourier transform in $G(3, 0, 0)$ applied pointwise over each real coefficient of $Q$ and defined as:

$$\hat{Q}(\xi) = \mathcal{F}\{Q\}(\xi) = \hat{Q}_0 + \hat{Q}_1 e_1 + \hat{Q}_2 e_2 + \hat{Q}_3 e_3 + \hat{Q}_{12} e_{12} + \hat{Q}_{13} e_{13} + \hat{Q}_{23} e_{23} + \hat{Q}_{123} e_{123}. \tag{7}$$

$Q$ is defined as the sum of all the multivectors in output of the Geometry blocks. The codomain of $\mathbf{\Phi}(Q)$ is also multivector valued, and each multivector in output of the 3D Clifford FNO, which we refer to $Q'$, is defined on a grid with the same resolution $M$ of the inputs. The FNO captures global interactions within the geometry and maps the input multivectors from a geometrical to a physics domain.

**3D Clifford Physics block.** The 3D Clifford Physics block is analogous to its Geometry counterpart. It differs from it since brings the $C$ channels of multivector $Q'$ in output of the FNO down to 1. As we estimate two different quantities, we have two different blocks to output $P'$ and $V'$, the pressure physics multivector and velocity physics multivector, respectively, for which we set

$$\langle P' \rangle_0 = \phi \tag{8}$$

Table 1: The three steps of the Fengbo pipeline.

| Module | Input | Output | Purpose |
|---|---|---|---|
| 1. 3D Clifford Geometry blocks | $\mathbf{P} = \{P_i\}_{i=1}^{N_g}$ | $\mathbf{Q} = \{Q_i\}_{i=1}^{N_g}$ | local, upsample, grade mixing |
| 2. 3D Clifford FNO | $Q = \sum_{i=1}^{N_g} Q_i$ | $Q'$ | global, PDE modelling |
| 3. 3D Clifford Physics blocks | $Q'$ | $\mathbf{P}' = \{P'_i\}_{i=1}^{N_p}$ | local, downsample, grade mixing |

$$\langle V' \rangle_1 = \psi_1 e_1 + \psi_2 e_2 + \psi_3 e_3 \tag{9}$$

where $\langle \cdot \rangle_k$ is the grade projector operator, which extracts the $k$-grade element out of the multivector. In short, Fengbo models the PDE solution problem as a mapping $\Xi(\cdot)$ of 3D (geometry) multivectors onto 3D (physics) multivectors in 3D Clifford Algebra $G(3, 0, 0)$, i.e.

$$\mathbf{P}' = \Xi(\mathbf{P}) \tag{10}$$

in which $\mathbf{P} = \{P_i\}_{i=1}^{N_g}$, with $N_g$ the number of input geometries in the dataset, and $\mathbf{P}' = \{P'_i\}_{i=1}^{N_p}$, with $N_p$ the number of output physical quantities to estimate. $N_g$ and $N_p$ determine the number of Geometry and Physics blocks in Fengbo, respectively. The steps in the Fengbo architecture are summarised in Table 1. Additional insight on each block is provided in Appendix B.

## 4 EXPERIMENTS

### 4.1 DATASETS

**ShapeNet Car.** The *ShapeNet Car* dataset is a subset of the larger ShapeNet 3D model repository consisting of thousands of realistic 3D car models employed in a CFD simulation with constant inlet flow velocity. It contains 500 shapes for training and 111 for testing. For this dataset, $N_g = 2, N_p = 2$, i.e. $\mathbf{P} = \{P, V\}$ (two inputs geometries) and $\mathbf{P}' = \{P', V'\}$ (two physical quantities to estimate, defined over the two different geometries).

**Ahmed Body.** The *Ahmed Body* dataset consists of CFD simulations with varying inlet flow velocity $\psi_{in}$. It contains 500 parametric variations of CFD simulations over simplified car models for training and 51 for testing. For this dataset, $N_g = 1, N_p = 1$, i.e. $\mathbf{P} = \{P\}$ (a single input geometry) and $\mathbf{P}' = \{P'\}$ (one physical quantity to estimate, no velocity field information provided). The inlet velocity is a crucial component since the output pressure field range depends on it. We embedded it as the trivector component of $P$ since it has a single component in one direction, i.e. $\psi_{in} e_1 + 0 e_2 + 0 e_3$. We do so by setting $\langle P \rangle_3 = (m_p \psi_{in}) e_{123}$, in which $m_p$ is the binary mask.

### 4.2 METRICS

We assess the quality of the pressure and velocity fields estimation through relative L2 norm (a percentage), defined as:

$$\mathcal{L}_P = \frac{\|\langle P'_{GT} \rangle_0 - \langle P' \rangle_0\|_2}{\|\langle P'_{GT} \rangle_0\|_2} = \frac{\|\phi(\mathbf{x}) - \hat{\phi}(\mathbf{x})\|_2}{\|\phi(\mathbf{x})\|_2} \tag{11}$$

$$\mathcal{L}_V = \frac{\|\langle V'_{GT} \rangle_1 - \langle V' \rangle_1\|_2}{\|\langle V'_{GT} \rangle_1\|_2} = \frac{\|\psi(\mathbf{x}) - \hat{\psi}(\mathbf{x})\|_2}{\|\psi(\mathbf{x})\|_2} \tag{12}$$

in which $\hat{\phi}, \hat{\psi}$ represent estimated pressure and velocity fields via Fengbo, extracted as the grade-0 and grade-1 component of estimated physics multivectors $P', V'$, respectively, while $\phi, \psi$ represent ground truth fields, extracted as the grade-0 and grade-1 component of ground truth physics multivectors $P'_{GT}, V'_{GT}$, respectively. The relative L2 norm has also been employed as loss function during training. Training details are discussed in detail in Appendix C. Code scripts can be found here, while trained model weights are available upon request.

### 4.3 RESULTS

Results are summarised in Tables 2-3 for the *ShapeNet Car* and the *Ahmed Body* datasets, respectively. Fengbo outperforms all variants of vanilla Fourier and Graph Neural Operators, as well as

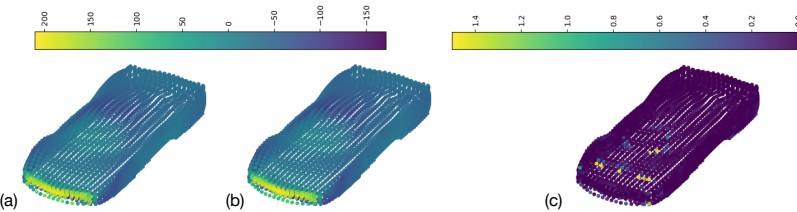

Figure 4: (a) Ground truth pressure field (b) Fengbo's estimated pressure field and (c) their relative error for a test shape in the *ShapeNet Car* dataset.

Table 2: Training and testing errors in pressure and velocity prediction on the *ShapeNet Car* dataset. Fengbo results have been obtained with $\alpha, \beta = \{5, 1\}$ for pressure and with $\alpha, \beta = \{1, 50\}$ for velocity, see Appendix C.

| Model | Pressure | | Velocity | |
|---|---|---|---|---|
| | training error | testing error | training error | testing error |
| MLP | - | 13.0 | - | 5.12 |
| PointNet Qi et al. (2017a) | - | 11.0 | - | 4.94 |
| Graph U-Net Gao & Ji (2019) | - | 11.0 | - | 4.71 |
| GraphSage Hamilton et al. (2017) | - | 10.5 | - | 4.61 |
| MeshGraph Net Pfaff et al. (2020) | - | 7.81 | - | 3.54 |
| GNO Li et al. (2020b) | 18.2 | 18.8 | - | 3.83 |
| Geo-FNO Li et al. (2020a) | 10.8 | 15.9 | - | 16.7 |
| UNet Ronneberger et al. (2015) | 12.5 | 12.8 | - | - |
| FNO Li et al. (2020a) | 9.65 | 9.42 | - | - |
| GINO (encoder-decoder) Li et al. (2024) | 7.95 | 9.47 | - | 3.86 |
| GINO (decoder) Li et al. (2024) | 6.37 | 7.12 | - | - |
| Fengbo [ours] | 6.94 | 8.86 | 3.23 | 3.47 |

Table 3: Training and testing errors in pressure prediction on the *Ahmed Body* dataset.

| | M.Gr.Net | UNet | FNO | GINO (e-d) | GINO (d) | GINO (e-d), $r=0.025$ | GINO (d), $r=0.025$ | GINO (e-d) $r=0.035$ | GINO (d) $r=0.035$ | Fengbo [ours] |
|---|---|---|---|---|---|---|---|---|---|---|
| **training** | 9.08 | 9.93 | 13.0 | 9.36 | 9.34 | 12.9 | 12.6 | 9.26 | 8.82 | 8.00 |
| **testing** | 13.9 | 11.2 | 12.6 | 9.01 | 8.31 | 12.8 | 12.7 | 9.30 | 9.39 | 10.7 |

UNet and Mesh GraphNet, and it yields comparable results to GINO. For the *ShapeNet Car* dataset, for example, Fengbo is able to estimate the pressure field with a 0.6% lower relative L2 norm compared to the GINO in its encoder-decoder (e-d) configuration, but with a 1.6% higher error compared to its decoder-only (e) configuration. Fengbo does so, however, with 60% fewer trainable parameters, with a reduced computational cost which does not depend on the degree of a graph representation of the input data, and being the only architecture reported able to do so while jointly estimating the scalar pressure field and the 3D velocity vector field. Fengbo achieves competitive accuracy thanks to this coupling of physical quantities, which allows to do so through simple convolutions on coarsely discretised meshes. This is especially notable when compared to more sophisticated architectures, including Geo-FNO Li et al. (2023), which degenerates when dealing with complex geometry, as shown in Li et al. (2024); Wu et al. (2024), despite it being precisely designed to learn to deform irregular domains onto a regular grid to be fed into the FNO. Models such as ONO Xiao et al. (2023) and OFormer Li et al. (2023), which are transformer-based, also become unstable when dealing with large meshes, as found in Wu et al. (2024).

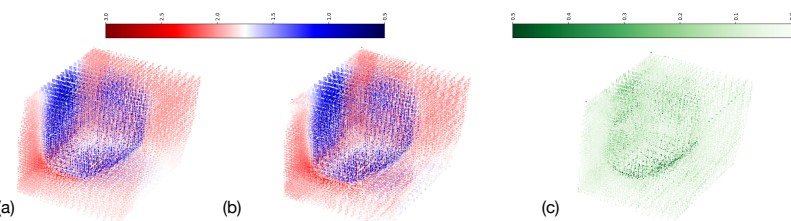

Figure 5: (a) Ground truth velocity field (b) Fengbo's estimated velocity field and (c) their relative error for a test shape in the *ShapeNet Car* dataset.

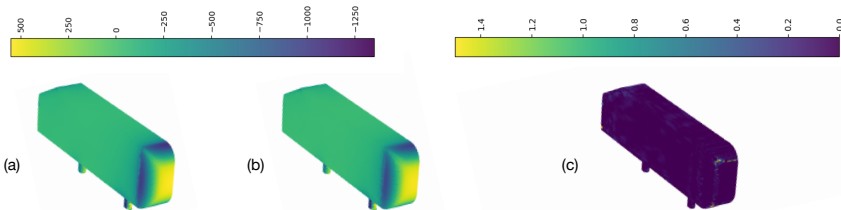

Figure 6: (a) Ground truth pressure field (b) Fengbo's estimated pressure field and (c) their relative error for a test shape in the *Ahmed Body* dataset.

Interestingly, estimating the velocity vector field appears to be an easier task to tackle. This is likely due to the significantly smaller variance of the velocity data as opposed to the sharp pressure variation over the car surface, as well as the fact that the velocity field $\psi(\mathbf{x})$ is defined over a point cloud containing $10\times$ more points as opposed to $\phi(\mathbf{x})$. This is mirrored also in the smaller gap between training and testing errors. The additional estimation of the velocity vector field does not imply a computational overhead, since the vector and pressure fields exist naturally within the multivector-based formulation of the problem and they are both embedded in a fixed-size volume. Note that this would not apply to graph-based methods, in which a larger cell count would mean a larger number of nodes, increasing drastically the computational complexity (see Table 4).

Similar observations can be made for the *Ahmed Body* dataset. Fengbo outperforms all models reported, with the exception of some GINO configurations depending on the choice of the radius of the Graph Neural Operator module. It is worth mentioning that experiments in Li et al. (2024) could benefit from the joint estimate of the wall shear stress, a physical parameter which was missing in the version of the dataset we employed. We are positive that regressing *also* on wall shear stress in a joint fashion, just like pressure and velocity for *ShapeNet Car*, could bring down the error of $10.7\%$ we obtained on the test set with Fengbo. Note also how Fengbo attains a training error of just $8\%$, the lowest out of every other model reported, indicating how additional parameter optimisation could be performed and likely reduce overfitting to bring the error down even further.

Comparison of estimates with Fengbo and corresponding ground truth pressure fields are given in Figs 4 and in Fig. 6 for the *ShapeNet Car* and *Ahmed Body* datasets, respectively. Note how errors in the pressure field are generally isolated points in a more or less uniform region with relative error close to zero. We are convinced that by simply smoothing the predicted field we could mitigate this issue and improve performance.

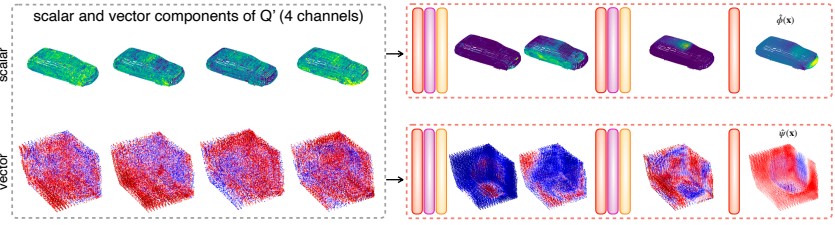

Figure 7: Intermediate outputs are interpretable physical quantities: $Q'$, the output of the 3D Clifford FNO block, is processed by $N_p = 2$ 3D Physics blocks in parallel to obtain $\hat{\phi}(\mathbf{x})$ and $\hat{\psi}(\mathbf{x})$, the pressure and velocity fields, respectively.

In Fig. 5, the ground truth and estimated velocity fields for a test case in the *ShapeNet Car* are reported. Note how the range of the relative L2 error in Fig. 5c is significantly smaller than the ranges in Figs. 4c - 6c. This is likely due to the denser, larger point clouds over which $\psi(\mathbf{x})$ is defined. Note also how larger errors are concentrated in the areas surrounding the outline of the car. Small discontinuities in the estimated field with respect to ground truth can be noticed in Fig. 5b, for example in the bottom right section: just like for $\hat{\phi}(\mathbf{x})$, we believe that smoothing the estimated field $\hat{\psi}(\mathbf{x})$ can further reduce the prediction error.

An example of the interpretability offered by Fengbo is given in Fig. 7. $Q'$, the multivector in output of the 3D Clifford FNO module, is processed by the the 2 Clifford Physics blocks to obtain 2 multivector $P'$ and $V'$, of which we extract the scalar part $\hat{\phi}(\mathbf{x})$ and the vector part $\hat{\psi}(\mathbf{x})$, respectively. Note that we are still dealing with full grade multivectors defined over the entire domain $\mathcal{D}$, but for the sake of visualisation we only plot the scalar and vector component masked by $m_s$ and $m_v$, respectively. As the velocity and pressure field are processed, is it possible to have a visual intuition on how they are being transformed into the final estimate. As the quantities plotted are scalars and vectors throughout, they carry physical meaning and cannot be interpreted as anything else but pressure and velocity fields, therefore we can claim that Fengbo is a whitebox model. This concept of interpretable convergence is analogous to that presented in Pepe et al. (2024a) for protein structures and in Pepe et al. (2024d) for camera poses.

Table 4: Comparison of different models. $d$ is the maximum degree of the graph, $D$ is the feature space dimensionality. *: See Appendix E.

| Model | Range | Complexity | Irregular Grid | Discretisation Convergent |
|---|---|---|---|---|
| PointNet Qi et al. (2017a) | global | $\mathcal{O}(N)$ | ✓ | ✗ |
| PointNet++ Qi et al. (2017a) | local-global | $\mathcal{O}(N \log N)$ | ✓ | ✗ |
| GNN Scarselli et al. (2008) | local | $\mathcal{O}(Nd)$ | ✓ | ✗ |
| CNN LeCun et al. (1995) | local | $\mathcal{O}(N)$ | ✗ | ✗ |
| UNet Ronneberger et al. (2015) | global | $\mathcal{O}(N)$ | ✗ | ✗ |
| Transformers Vaswani (2017) | radius $r$ | $\mathcal{O}(N^2)$ | ✓ | ✓ |
| Transolver Wu et al. (2024) | local-global | $\mathcal{O}(NSC + NS^2)$* | ✓ | ✓ |
| Galerkin Cao (2021) | global | $\mathcal{O}(ND^2)$ | ✓ | ✓ |
| MeshGraphNet Pfaff et al. (2020) | local-global | $\mathcal{O}(Nd)$ | ✓ | ✓ |
| GNO Li et al. (2020b) | global | $\mathcal{O}(Nd)$ | ✓ | ✓ |
| FNO Li et al. (2020a) | global | $\mathcal{O}(N \log N)$ | ✗ | ✓ |
| Geo-FNO Li et al. (2020a) | global | $\mathcal{O}(N \log N)$ | ✓ | ✓ |
| GINO Li et al. (2024) | local-global | $\mathcal{O}(N \log N + Nd)$ | ✓ | ✓ |
| Fengbo [ours] | local-global | $\mathcal{O}(N \log N)$ | ✗ | ✓ |

g

Fengbo has a computational complexity of $\mathcal{O}(N \log N)$ (see Table 4): the embedding into a 3D volume has complexity $\mathcal{O}(N)$, and the limiting component on the computational complexity is given by the Clifford FNO module, with complexity $\mathcal{O}(N \log N)$. Moreover, Fengbo's accuracy is minimally impacted by smaller grid resolutions, making it robust to coarses discretisations and hence discretisation convergent (see Ablation Study in Appendix D).

## 5 CONCLUSIONS

We introduced Fengbo, a Neural Operator pipeline able to solve large-scale, 3D PDEs over complex shapes which sits entirely in 3D Clifford Algebra. With Fengbo, we combine the descriptive power of Neural Operators with the inductive bias and interpretability of networks in Clifford Algebra to obtain a compact pipeline that is able to estimate multiple physical quantities both accurately and at once, without extra computational overhead.

We reported results on the two 3D CFD datasets available, *ShapeNet Car* and *Ahmed Body*, yielding a test error lower than most previously reported NO models. We are able to do so with a model with about $60\%$ fewer parameters and with reduced computational complexity compared to graph- and transformer-based models. Moreover, we also estimate jointly the velocity field, leveraging exclusively geometrical information of $\Omega_D$. Fengbo is thus a lightweight, expressive and accurate pipeline entirely in 3D Euclidean space.

**Limitations**. Due to limited computational resources, we could not test deeper, larger versions of our proposed architecture, which caps at 42 million parameters. Fengbo might struggle when applied on less informative datasets (e.g. fewer physical variables, lack of geometrical information) that do not allow for the construction of full-grade multivectors. Besides, most implementations of Clifford Algebra networks rely on a tensor representation of multivectors, negatively impacting the model training speed.

**Future work**. Future work might include testing a larger Fengbo over multiple datasets, projecting it down to 2D for or extending its use for different PDEs to estimate jointly multiple physical quantities defined over complex, irregular domains.

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

# A    CLIFFORD ALGEBRA FUNDAMENTALS

## A.1    DEFINING A SUBALGEBRA

A Clifford algebra of size $n$ can be defined over a scalar field and a set of $n$ independent basis vectors $\{e_i\}_{i=1,...n}$. We indicate a generic closed subalgebra with $G(p,q,r)$ or alternatively $G(p,q,r)$, with $n = p + q + r$. A closed subalgebra $G(p,q,r)$ has $p$ basis vectors that square to 1, $q$ basis vectors that square to -1 and $r$ basis vectors that square to 0.

## A.2    THE GEOMETRIC PRODUCT

Elements in a GA are called $multivectors$. Elements of any type can be added or multiplied together. Each element has a $grade$ associated with it. By grade we define the dimension of the hyperplane an object specifies. e.g. scalars are grade 0, vectors are grade 1, bivectors are grade 2, etc. Clifford algebra is also known as Geometric algebra because of the geometric product. The geometric product between two vectors is given by

$$ab = a \cdot b + a \wedge b \tag{13}$$

in which the scalar (or $inner$) product $a \cdot b$ is the usual scalar product of linear algebra equal to the cosine of the angle between $a$ and $b$, while the wedge (or $outer$) product $a \wedge b$ produces a bivector (e.g. an oriented plane). The geometric product between vectors is hence the sum of a scalar and a bivector, that have different grade. Any multivector of a unique grade $r$ that can be defined as $A = a_1 \wedge a_2 \wedge ... \wedge a_r$ is called a blade.

The reversion operator for a multivector is given by $\tilde{A}$. The reverse of a scalar is equal to the scalar itself and the reverse of a vector is equal to the vector itself. For a multivector, we have that

$$(AB)\tilde{} = \tilde{B}\tilde{A}$$
$$(A + B)\tilde{} = \tilde{A} + \tilde{B} \tag{14}$$

The general rule to reverse a $r$-blade is given by

$$\tilde{A}_r = (-1)^{\frac{r(r-1)}{2}} A_r \tag{15}$$

The geometric product between a multivector and its reverse gives the squared magnitude: $|A|^2 = \langle A\tilde{A} \rangle_0$. The reversion operator can be used to define the inverse of a multivector as

$$A^{-1} = \frac{\tilde{A}}{|A|^2} \tag{16}$$

It can be easily shown that $A^{-1}A = 1$ when $A\tilde{A}$ is a scalar.

The grade projector operator is denoted by $\langle A \rangle_r$, where $r$ is the grade we want to extract from $A$. This comes from the fact that a multivector in an $n$ dimensional algebra can be written as

$$A = \sum_{i=0}^{n} \langle A \rangle_i \tag{17}$$

The dual of a multivector is defined as

$$A^* = AI_n^{-1} \tag{18}$$

where $I_n$ is called the $pseudoscalar$, defined as $I_n = e_1 \wedge e_2 \wedge ... \wedge e_n$. The pseudoscalar is the highest grade element in a GA. The product of grade-$n$ pseudoscalar $I_n$ and grade-$r$ multivector $A_r$ is a grade-$(n - r)$ multivector, and is termed the duality transformation. The pseudoscalar interchanges inner and outer products:

$$A_r \cdot (B_s I_n) = \langle A_r B_s I_n \rangle_{|r-(n-s)|} = \langle A_r B_s I_n \rangle_{n-(r+s)} = \langle A_r B_s \rangle_{r+s} I_n = A_r \wedge B_s I_n \tag{19}$$

### A.3 Clifford Algebra of the plane and of space

The Clifford algebra of the Euclidean plane is denoted by $G(2,0,0)$, with two basis vectors $e_1, e_2$. Being an $n = 2$ dimensional GA (since $p + q + r = 2 + 0 + 0 = 2 = n$), it is spanned by $2^2 = 4$ elements, namely a scalar, two vectors $e_1, e_2$ and the bivector $e_1 e_2 = e_1 \wedge e_2$. $G(2,0,0)$ includes the concept of complex numbers, since the pseudoscalar of this algebra $I = e_1 \wedge e_2 = e_1 e_2 = e_{12}$ squares to -1, since $I^2 = e_{12}^2 = e_{12} e_{12} = (e_1 e_2)(e_1 e_2) = -(e_1 e_1)(e_2 e_2) = -1$. A scalar plus a bivector can be seen as a representation of a complex number, since $Z = a + Ib \equiv a + \iota b$, where $\iota$ is the imaginary unit. Similarly, if $X = ae_1 + be_2 = e_1(a + bI) = e_1 Z$.

Adding a third basis vector $e_3$ we form $G(3,0,0)$, the GA of Euclidean space, which is what we employed in this paper. It has $2^3 = 8$ elements, a scalar, three vectors, three bivectors $(e_{12}, e_{23}, e_{13})$ and one *trivector* $(e_{123} = e_1 \wedge e_2 \wedge e_3 = I_3$, the pseudoscalar). The GA of space includes quaternion algebra, since a quaternion $\mathbf{q} = w + a\mathbf{i} + b\mathbf{j} + c\mathbf{k}$ can be represented as a multivector $A = w + ae_{12} + be_{13} + ce_{23}$.

## B Implementation Details

Details of the Fengbo architecture are shown in Fig. 8. The 3D Clifford Geometry block (Fig. 8a) consists of 3 3D convolutional layers with kernel size $5 \times 5 \times 5$. The first 2 convolutions are followed by a group normalisation layer and a GeLU activation function. The block yields multivectors with progressively increasing number of channels $C_g = \{1, 2, 4\}$.

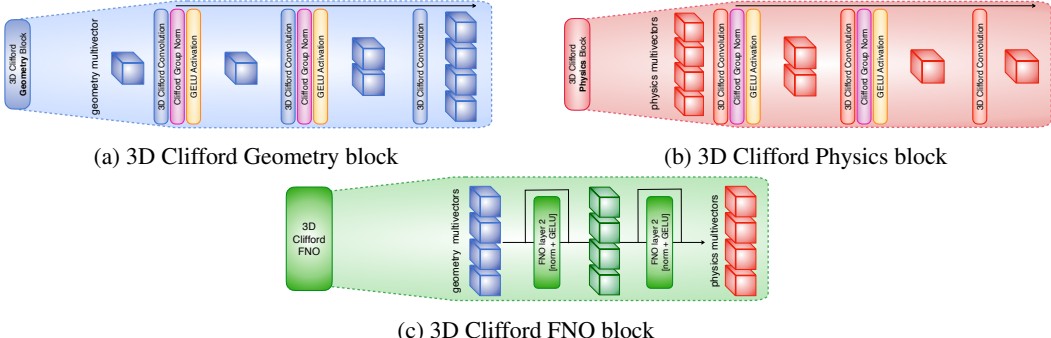

(a) 3D Clifford Geometry block    (b) 3D Clifford Physics block

(c) 3D Clifford FNO block

Figure 8: The three components of the Fengbo architecture.

An example of the geometric interpretability of the intermediate outputs of Fengbo is given in Fig. 9. Multivector $P$, containing a scalar, vector and bivector component, is processed by the 3D Clifford Geometry block to obtain the 4-channel-multivector $Q$. We employ a grayscale colormap for scalar quantities, i.e. scalar and trivector components, and $jet$ and $rainbow$ colormaps for the vector and bivector components, respectively. The input shape multivector $P$ built from the dataset is scattered within the $[-1, 1]$ volume, bounded by the $tanh$ activation function. Elements of different grades are mixed, as can be noticed from the appearance of trivector components. The sequence of convolutions makes the 3D multivectors progressively denser. The last multivector $Q_P$, with 4-channels, is unbounded due to the lack of an activation function and fed into the 3D Clifford FNO. Each channel shows how different grade elements in the volume cluster to form different shapes, more or less aligned in a certain direction. While far from the original car shape, these *blobs* indeed represent scalar, vectors, bivectors and trivectors in 3D space: the vector components shown, for example, cannot be interpreted as anything else than coordinates of 3D point clouds precisely because of our choice of embedding. Similar considerations can be made for $V$ and $Q_V$.

Fig 8b shows the 3D Clifford Physics block. It contains the same layers as the 3D Clifford Geometry block, but with a decreasing number of channels $C_p = \{2, 1, 1\}$, as shown in Fig. 7, and a different meaning attached to the multivector representation, where the scalar and vector part represent the pressure field $\phi(\mathbf{x})$ and the velocity field $\psi(\mathbf{x})$.

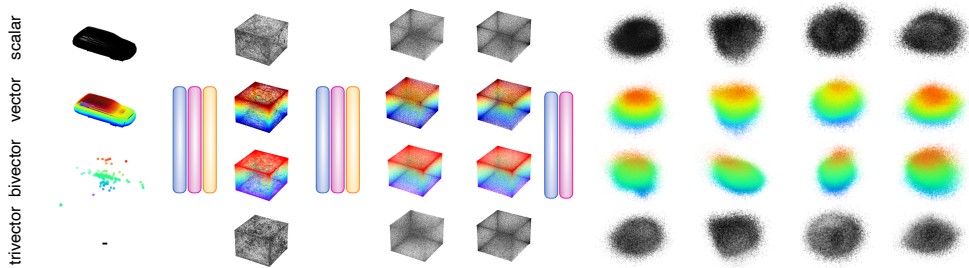

Figure 9: Intermediate layers outputs from $P$ to $Q_P$ within the 3D Clifford Geometry block for a test sample in the *ShapeNet Car* dataset.

Fig 8c shows the 3D Clifford FNO block. It includes $F = 2$ residual blocks, 4 input channels, 4 output channels, $C = 25$ hidden channels and $m = 8$ Fourier modes for each of its 3 dimensions. Each layer is followed by a group normalisation layer and a GeLU activation function.

**Pipeline Generalisability**. Fengbo operates entirely within Clifford Algebra, which is naturally suited for a multidimensional treatment Lasenby & Lasenby (2001); Lasenby & Doran (2001); Doran & Lasenby (2003); Hitzer (2012), and extensions of a same architecture in Clifford Algebra to higher dimensions has been widely documented in the literature as one of the main advantages of these types of networks Brandstetter et al. (2022); Ruhe et al. (2024; 2023); Pepe et al. (2024c). The implementation of a 2D equivalent to our 3D Fengbo pipeline is straightforward, as detailed in Table 5. By projecting the convolutional operations from 3D to 2D and operating within either $G(2, 0, 0)$ or $G(0, 2, 0)$, we achieve a fully analogous approach for 2D problems with targets that include scalar fields or 2D vector fields. Besides, just like Fengbo 3D is not limited to the task of 3D flow estimation, Fengbo 2D is not limited to 2D flows, but it can be extended to any 2D PDE that establishes a mapping from the geometry to the physics of the problem.

Table 5: Comparison of Fengbo's 3D and 2D configurations.

| Model | Algebra | Dimensionality ($D$) | Tensor Shape | Complexity | Geometry Block | FNO | Physics Block | Normalisation | Targets |
|---|---|---|---|---|---|---|---|---|---|
| Fengbo 3D | $G(3,0,0), G(0,3,0)$ | 8 | $C \times M \times M \times M \times D$ | $\mathcal{O}(N \log N), N = M^3$ | 3D Clifford Convolutions | 3D full-grade Spectral Convolutions, 3 Fourier modes | 3D Clifford Convolutions | Group Normalisation 3D | 2 scalar fields, 1 3D vector field, 1 3D bivector field. |
| Fengbo 2D | $G(2,0,0), G(0,2,0)$ | 4 | $C \times M \times M \times D$ | $\mathcal{O}(N \log N), N = M^2$ | 2D Clifford Convolutions | 2D full-grade Spectral Convolutions, 2 Fourier modes | 2D Clifford Convolutions | Group Normalisation 2D | 2 scalar fields, 1 2D vector field. |

We can consider the 2D Fengbo as a simpler subcase of the 3D case, since its 3D implementation presents several more challenges, namely:

- 6-dimensional tensors in 3D, with shape $B \times C \times M \times M \times M \times D$, where $B$ is the batch size, $C$ is the number of channels, $M$ is the grid resolution and $D$ is the algebra dimensionality as opposed to 5-dimensional tensors in 2D, with shape $B \times C \times M \times M \times D$, which require significantly less memory and allow for larger model sizes.

- Larger algebra dimensionality, $D = 2^3 = 8$ elements in $G(3, 0, 0)$, namely 1 scalar, 1 trivector, 3 vectors and 3 bivectors, as opposed to the 2D case with $D = 4$, with 1 scalar, 1 bivector and 2 vectors. This has implications in the sparsity of the input tensors and in their memory requirements, which negatively impact convergence.

- Significantly higher computational complexity, since it stays the same for both cases, namely $\mathcal{O}(N \log N)$, but with $N = M^3$ for the 3D case and $N = M^2$ for the 2D case. Moving to 2D would allow for a larger discretisation that can preserve a higher level of detail at a fraction of the computational cost.

Moreover, handling 3D datasets with the proposed pipeline inherently includes the capability to process 2D datasets of Li et al. (2020a); Wu et al. (2024). This is shown in Table 6. When processing instances in *ShapeNet Car* and *Ahmed Body*, we: 1) sample points from the unstructured meshes; 2) discretise the irregular point clouds onto regular grids and 3) embed them in multivector form. Datasets like *AirFRANS*, which also contains unstructured meshes, would be processed in the same way. All the remaining 6 datasets fall into data structures which are intermediate steps of the pipeline we established with Fengbo: point clouds, like the *Elasticity* dataset, can be directly discretised onto

regular grids and embedded into multivector form, regular grids of the *Plasticity, Airfoil and Pipe* datastes can be simply embedded as multivectors, as already demonstrated in Brandstetter et al. (2022); Pepe et al. (2024b), and structured meshes of the *Navier-Stokes* and *Darcy* datasets can be processed like their unstructured counterparts, demonstrating Fengbo's generalisability to 2D cases.

Table 6: Data representations and their processing steps with the Fengbo pipeline.

| | Unstructured Meshes | Point Clouds | Regular Grid | Structured Mesh |
|---|---|---|---|---|
| Datasets | *ShapeNet Car, Ahmed Body* (3D), *AirFRANS* (2D) | *Elasticity* (2D) | *Plasticity, Airfoil, Pipe* (2D) | *Navier-Stokes, Darcy* (2D) |
| 1. Sample points from mesh | ✓ | - | - | ✓ |
| 2. Discretise onto regular grid | ✓ | ✓ | - | ✓ |
| 3. Embed in multivector form | ✓ | ✓ | ✓ | ✓ |

## C  TRAINING DETAILS

Fengbo was trained on 3 NVIDIA A100 GPUs with 40GB RAM. It was trained for 100 epochs with a batch size of 3, for a total of approximately 24 compute hours. We employed the Adam optimiser to update the model's weights with default parameters of $\beta_1 = 0.9, \beta_2 = 0.999$. We adopted a learning rate of $10^{-4}$, reduced on plateau by a factor of 2 with patience on the validation loss set to 8 epochs. The loss we minimised for the *ShapeNet Car* dataset is

$$\mathcal{L} = \alpha \mathcal{L}_P + \beta \mathcal{L}_V + \|\phi - \hat{\phi}\|_1 \tag{20}$$

with parameters $\alpha = 5, \beta = 1$ picked empirically to weight the pressure component more, since it was harder to regress. A similar loss was employed for the *Ahmed Body* dataset:

$$\mathcal{L} = \alpha \mathcal{L}_P + \beta \|r - \hat{r}\|_1 + \|\phi - \hat{\phi}\|_1 \tag{21}$$

but with $\alpha = 1, \beta = 1$ and $r$ being the Reynolds number embedded as the trivector component of the output, i.e. $\langle R' \rangle_3 = (m_p r) e_{123}$, where $R'$ refers to the second output of the network, $P'$ being the first. No velocity information is provided. The L1 norm term on pressure was added to further penalise large deviations of $\hat{\phi}$, the estimated pressure field with respect to ground truth.

We employed elastic net regularisation, with $\lambda_1 = 10^{-5}$ lasso regularisation coefficient and $\lambda_2 = 10^{-4}$ ridge regularisation coefficient, to encourage group selection of correlated features and reduce overfitting, which we found to be significant over such small training sets. For training, input geometries are normalised in the range $[-1, 1]$, vector velocity fields are normalised in the range $[0, 1]$ and pressure fields are unit normalised by subtracting their mean and dividing them by their variance. Test metrics are measured over denormalised quantities to yield physically meaningful errors.

## D  ABLATION STUDY

Table 7: Ablation on the impact of $M$.

| Grid Size $M$ | ShapeNet Car | | | | Ahmed Body | |
|---|---|---|---|---|---|---|
| | Pressure | | Velocity | | Pressure | |
| | training | testing | training | testing | training | testing |
| 40 | 6.69 | 12.8 | 5.58 | 6.11 | 10.6 | 15.1 |
| 50 | 7.38 | 11.2 | 5.99 | 5.70 | 10.4 | 13.9 |
| 60 | 6.42 | 10.1 | 5.17 | 5.26 | 9.17 | 12.3 |
| 70 | 7.43 | 10.5 | 5.97 | 5.61 | 8.10 | 11.7 |
| 80 | 6.94 | 8.86 | 5.56 | 5.10 | 8.00 | 10.7 |

**Pipeline Scalability**. We study the impact of four components of the Fengbo pipeline, namely the grid size $M$, the number of hidden channels in the 3D Clifford FNO module $C$, the number of blocks in the FNO $F$ and the number of modes in the FNO $m$.

The impact of grid size on Fengbo's accuracy is shown in Table 7. We train and test on volumes with the same resolution. It can be noted that, by reducing the number of voxels by $87.5\%$ (from $M = 80$ to $M = 40$), i.e. significantly reducing the level of details in our input shape, Fengbo still yields a test error only $4\%$ higher for the *ShapeNet Car* dataset and $4.5\%$ higher for the *Ahmed Body* dataset. We can hence claim discretisation convergence.

Since the 3D Clifford FNO is Fengbo's key component, we study the impact of its parameters. In Fig. 10 we report the ablations over the number channels $C$ within the FNO for the two datasets and for the relative $L_2$ norm over pressure and velocity fields $\phi, \psi$, respectively. For the velocity field of *ShapeNet Car* we tested two combinations of the weighting coefficients of the loss function $\alpha, \beta$. The grid size is fixed to $M = 80$ and the number of blocks within the FNO is fixed to $F = 2$, while we test $C = \{5, 10, 15, 20, 25\}$.

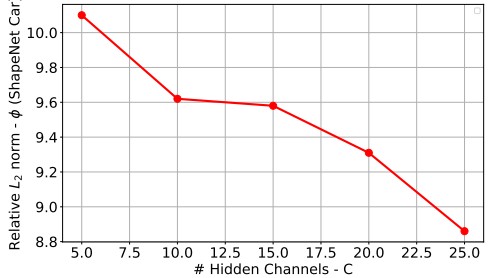
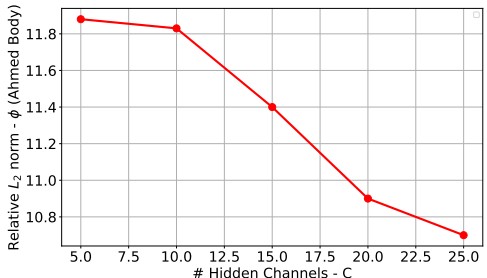

(a) Relative $L_2$ norm on pressure, *ShapeNet Car* dataset versus $C$.

(b) Relative $L_2$ norm on pressure, *Ahmed Body* dataset versus $C$.

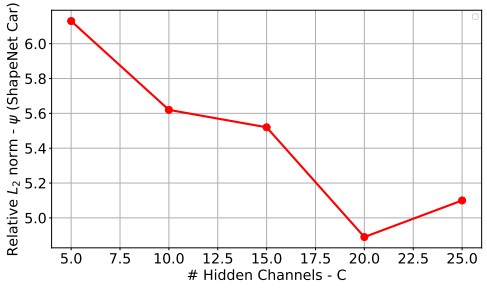
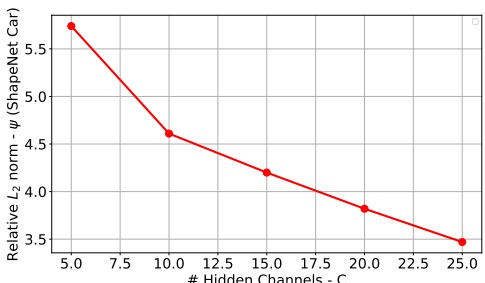

(c) Relative $L_2$ norm on velocity, *ShapeNet Car* dataset versus $C$. $\{\alpha, \beta\} = \{5, 1\}$

(d) Relative $L_2$ norm on velocity, *ShapeNet Car* dataset versus $C$. $\{\alpha, \beta\} = \{1, 50\}$

Figure 10: Ablation on the number of hidden channels $C$.

In all four presented scenarios, a higher number of hidden channels (i.e. a wider network) yields a steep decrease in the error at testing stage, proving how a larger version of Fengbo to that presented in the main body of this manuscript ($C = 25$) could further improve the quality of the PDE solution and demonstrating its scalability with respect to $C$.

In Fig. 11 we study the impact of the number of FNO blocks for the same four cases above. We tested $F = 1, 2, 3, 4$ by keeping $C = 15$ and $M = 80$. Also in this scenario, a deeper network corresponds to a more accurate estimation and hence scalability with respect to $F$. Note, in Fig. 11a, a lower absolute minimum for relative $L_2$ norm over $\phi$ for the *ShapeNet Car* dataset of $8.25\%$ with $F = 4$.

We then studied the impact of the number of Fourier modes $m$ of the FNO. We tested $m = \{3, 6, 8, 10, 12, 14\}$ by keeping $F = 2$ and $C = 20$. In this case results are less uniform across the four test cases, but we can conclude thar a larger number of modes often corresponds to similar if not worse performances, as already pointed out in Brandstetter et al. (2022).

The effect that these ablations have on the number of models parameters and size are reported in Fig. 13. Note how the ablations on $M$ are missing since they do not affect the model dimension. The model size scales exponentially with respect to $C, m$, and linearly with respect to $F$. Fourier

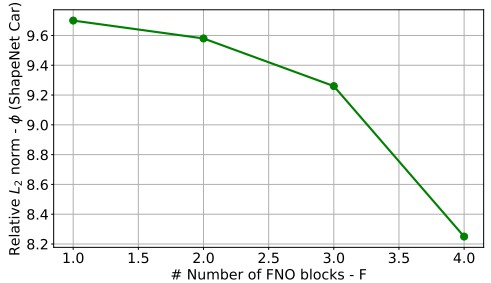

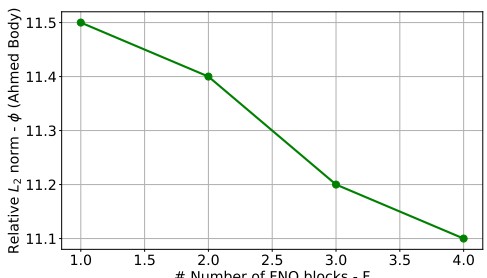

(a) Relative $L_2$ norm on pressure, *ShapeNet Car* dataset versus $F$.

(b) Relative $L_2$ norm on pressure, *Ahmed Body* dataset versus $F$. $\{\alpha, \beta\} = \{5, 1\}$.

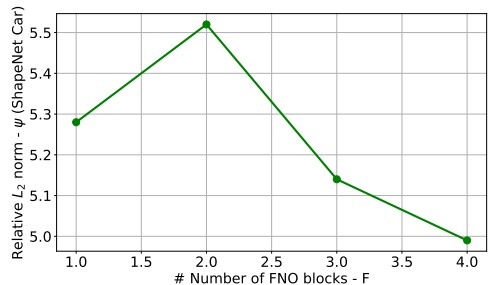

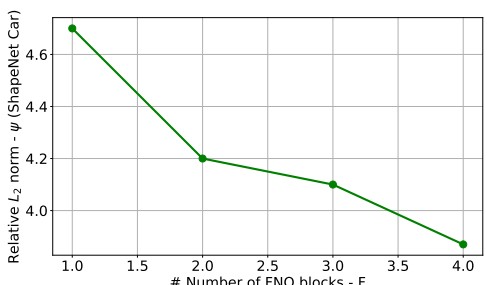

(c) Relative $L_2$ norm on velocity, *ShapeNet Car* dataset versus $F$.

(d) Relative $L_2$ norm on pressure, *Ahmed Body* dataset versus $F$. $\{\alpha, \beta\} = \{1, 50\}$.

Figure 11: Ablation on the number of FNO blocks $F$.

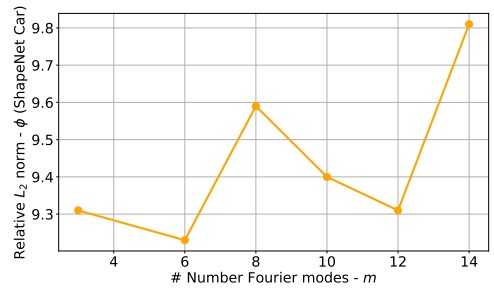

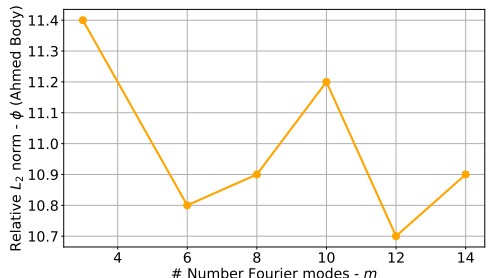

(a) Relative $L_2$ norm on pressure, *ShapeNet Car* dataset versus $m$.

(b) Relative $L_2$ norm on pressure, *Ahmed Body* dataset versus $m$.

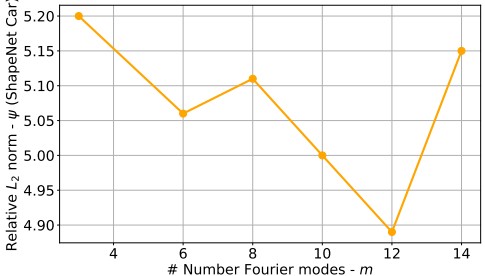

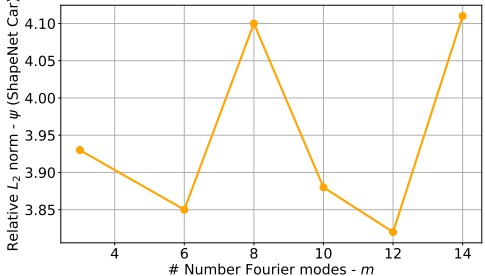

(c) Relative $L_2$ norm on velocity, *ShapeNet Car* dataset versus $m$. $\{\alpha, \beta\} = \{5, 1\}$.

(d) Relative $L_2$ norm on pressure, *Ahmed Body* dataset versus $F$. $\{\alpha, \beta\} = \{1, 50\}$.

Figure 12: Ablation on the number of Fourier modes $m$.

modes have the biggest impact on the model parameters, with $m = 14$ corresponding to a $\times 13$ increase with respect to the Fengbo presented in the main body of the text, without benefiting the test error. The number of blocks $F$, on the other hand, corresponds to a relatively milder increase in the model size while still providing a substantial improvement in performance. This proves our claim in the *Limitation* section, i.e. that a deeper network can likely correspond to more robust and accurate predictions than those shown in Table 2.

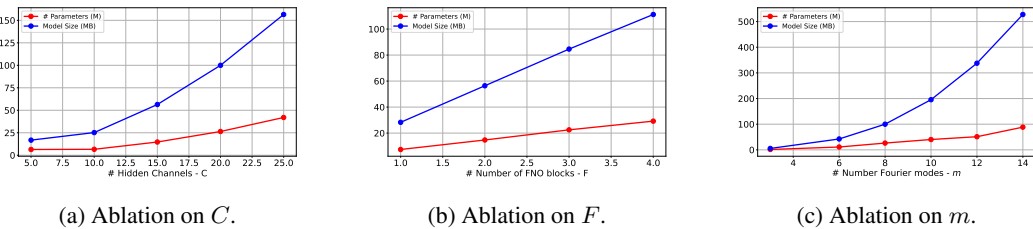

(a) Ablation on $C$.       (b) Ablation on $F$.       (c) Ablation on $m$.

Figure 13: Number of parameters (M) and model size (MB) as a function of $C, R, m$.

Lastly, the impact of the weighting coefficients of the loss function on Fengbo's accuracy is shown in Table 8. We fix the resolution to $M = 80$ and vary the weight attributed to different physical quantities in the loss function. $\beta$ weights velocity for the *ShapeNet Car* dataset and the Reynolds number for the *Ahmed Body* dataset, while $\alpha$ weights the pressure in both. Note how the Reynolds number does not contribute in a significant way to the estimation of pressure in the *Ahmed Body* dataset. From Table 8 we can conclude that the high accuracy accuracy of Fengbo stems also due to the joint estimation of variables, e.g regressing also on $\psi$ can better constraint the values of $\phi$ can assume and vice versa. $\alpha, \beta$ can be thought as two parameters that mix quantities to be regressed and that can be tuned based on the specific requirements of problem to be tackled, e.g. which quantity we wish to prioritise.

Table 8: Ablation on the impact of $\alpha, \beta$.

| $\alpha$ | $\beta$ | ShapeNet Car | | | | Ahmed Body | |
|---|---|---|---|---|---|---|---|
| | | Pressure | | Velocity | | Pressure | |
| | | training | testing | training | testing | training | testing |
| 1 | 0 | 8.53 | 9.21 | - | - | 8.60 | 11.8 |
| 1 | 1 | 9.07 | 9.32 | 7.23 | 4.39 | 8.00 | 10.7 |
| 2 | 1 | 8.03 | 9.30 | 6.40 | 4.56 | 8.23 | 10.9 |
| 5 | 1 | 6.94 | 8.86 | 5.56 | 5.10 | 7.64 | 10.9 |
| 10 | 1 | 5.38 | 9.12 | 4.28 | 5.48 | 9.31 | 11.9 |
| 0 | 1 | - | - | 4.90 | 4.03 | - | - |
| 1 | 2 | 9.37 | 9.50 | 4.09 | 4.12 | 8.42 | 11.9 |
| 1 | 5 | 7.71 | 9.83 | 3.98 | 3.82 | 8.34 | 11.8 |
| 1 | 10 | 9.37 | 10.1 | 3.82 | 3.60 | 8.26 | 11.8 |
| 1 | 20 | 10.5 | 10.8 | 3.37 | 3.59 | - | - |
| 1 | 50 | 8.42 | 11.4 | 3.23 | 3.47 | - | - |

# E  NOTES ON COMPUTATIONAL COMPLEXITY

As outlined in Section 2, we focused primarily on Neural Operators over Transformers due to the latter's significantly higher computational complexity, namely $\mathcal{O}(N^2)$. Transformers, albeit offering improved performances, introduce substantial challenges in terms of resource requirements and scalability. As a proof of that, we offer an analysis of the theorical complexity of the Fengbo model and compare it with the current state-of-the-art in Transformer-based solvers, the Transolver architecture Wu et al. (2024), which to the best of our knowledge is both the most accurate and the least computationally expensive Transformer architecture designed to solve PDEs. Albeit mostly validated over 2D problems, the Transolver has also been tested over one 3D dataset, namely *ShapeNet Car*.

More specifically, authors of Wu et al. (2024) designed an *ad hoc* attention layer, the Physics-Attention layer, that operates on slices of the PDE domain $\Omega_d$. The reported computational complexity of such layer $\mathcal{O}(NSC + S^2C)$, in which $N$ is the number of meshes, $S$ is the number of slices into which the domain is partitioned and $C$ is the number of hidden channels of the model. The authors claim a quasi-linear complexity with respect to $N$. However, the overall complexity is heavily dependent on the choice of the model parameters $S$ and $C$, and for large values of $S$ and $C$, which is the setting for most of the experiments in Wu et al. (2024), it becomes sub-quadratic. We compare the model complexities for the 3D and the 2D case.

**3D Case**. Model complexities for the 3D case are shown in Fig. 14. For the *ShapeNet Car*, specifically, the reported parameters are as follows: $N \simeq 32000$ meshes, $S = 32$ slices into which the car surface is partitioned and $C = 256$ channels. This places the computational complexity of the Transolver architecture for the *ShapeNet Car* dataset at the green marker shown in Fig. 14a.

On the other hand, as shown in Table 4, we report a computational complexity of $\mathcal{O}(N \log N)$ for the Fengbo pipeline. In our approach, $N$ is the dimension of the 3D regular grid, hence $N = M^3$, with $M$ being the grid resolution. This places the computational complexity of the Fengbo architecture for the *ShapeNet Car* dataset at the yellow marker shown in Fig. 14a.

Such design choices yielded results shown in Fig. 14b: the Transolver model attains a $1.4\%$ decrease relative $L_2$ error over Fengbo, but at a computational complexity of two orders of magnitude larger. It is also worth noting that the experiment setting for Transolver followed the implementation of 3D-GeoCA Deng et al. (2024), which takes 789 samples for training and 100 samples for testing. On the other hand, we followed the approach of GINO Li et al. (2024), in which we retain 611 watertight meshes and employ 500 samples for training and 100 for testing, meaning that the Transolver, besides being computationally more complex, was also trained on $57.8\%$ more samples.

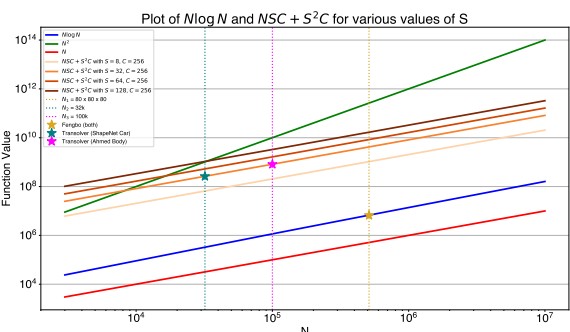

(a) Computational complexities of Fengbo ($N \log N$) and of Transolver ($NSC + S^2C$), for different values of $S$. Fengbo operates at $N_1$, while the Transolver operates at $N_2$ in its 3D setting. Linear and quadratic complexity curves added for reference.

|  | Fengbo | Transolver |
|---|---|---|
| Compl. | $\mathcal{O}(10^6)$ | $\mathcal{O}(10^8)$ |
| $L_2$ ($\phi$) | 8.86 | 7.45 |
| $L_2$ ($\psi$) | 3.47 | 2.07 |

(b) Comparison of Fengbo and Transolver in terms of computational complexity (yellow and green marker in the figure, respectively) and test $L_2$ norm for the pressure and velocity field over the *ShapeNet Car* dataset. Pressure $L_2$ obtained with $\{\alpha, \beta\} = \{5, 1\}$, Velocity $L_2$ obtained with $\{\alpha, \beta\} = \{1, 50\}$

Figure 14: Computational complexity comparison of Fengbo and Transolver Wu et al. (2024) for the 3D case.

A similar claim can be made for the *Ahmed Body* dataset, not analysed in Wu et al. (2024), but whose complexity can still be studied. Assuming one point per mesh, i.e. $N = 100000$, and the same parameters configuration employed for the *ShapeNet Car* dataset, i.e. $\{S, C\} = \{32, 256\}$, the resulting complexity of the model also reaches $\mathcal{O}(10^8)$, corresponding to to the pink marker in Fig. 14a. With Fengbo, the grid resolution is kept unchanged for the *Ahmed Body* dataset, which yields identical complexity to the *ShapeNet Car* dataset, demonstrating its robustness to larger mesh size.

**2D Case**. We compare the ablations presented in Appendix A of this manuscript with those presented in Appendix C of Wu et al. (2024). We compute the corresponding Fengbo complexity as the grid size varies, i.e. $M = \{40, 50, 60, 70, 80\}$ and the corresponding Transolver complexity as $N$ and $C$ vary across datasets and as the number of slices $S$ vary employed across ablations, namely $S = \{1, 8, 16, 32, 64, 96, 128, 256, 512, 1024\}$. This is summarised in Table 9. We then plot the

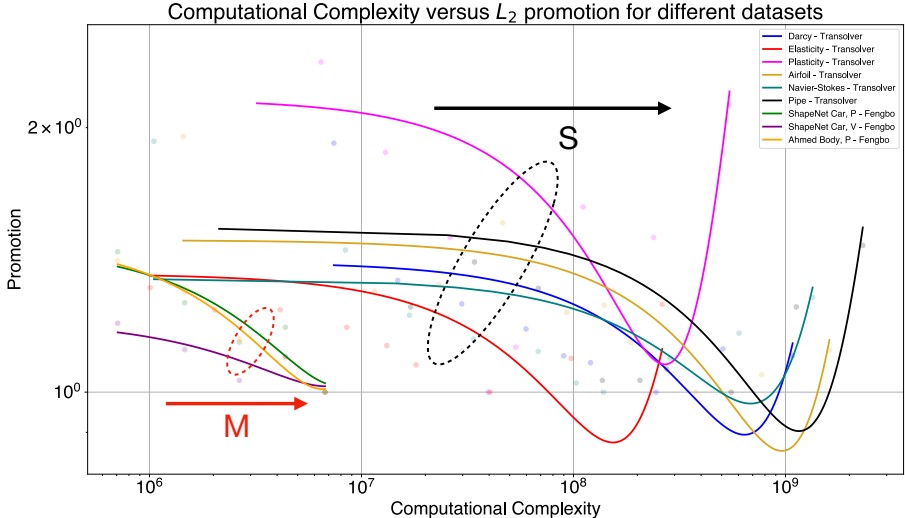

Figure 15: relative $L_2$ norm promotion versus computational complexity for Fengbo and Transolver's ablations.

relative $L_2$ norm promotion versus the resulting complexities for six 2D datasets (*Darcy*, *Elasticity*, *Plasticity*, *Airfoil*, *Navier-Stokes* and *Pipes*) and two 3D datasets (*ShapeNet Car* and *Ahmed Body*). The promotion is defined as $\mathcal{P}_i = \mathcal{L}_i/\mathcal{L}^*$, with $\mathcal{L}_i$ the relative $L_2$ norm reported for the $i$th ablation and $\mathcal{L}^*$ the overall minimum $L_2$ norm reported. We do so as different datasets might present very different ranges of $\mathcal{L}$.

This is shown in Fig. 15, in which the markers represent different ablations and the curves are the resulting interpolations. Note how, even when compared to 2D datasets, Fengbo still operates at one order of magnitude below Transolver for the *Elasticity* dataset and at two orders of magnitude for the remaining five datasets. On top of that, the computational cost to lower $\mathcal{L}$ to the optimal value is also significantly lower for the Fengbo pipeline. Additionally, if Fengbo were to be tested in a 2D scenario, the value of $N$ would likely be much smaller than $10^6$.

Table 9: Complexity comparison for different datasets

|  | Darcy | Elasticity | Plasticity | AirFoil | Navier Stokes | Pipes | ShapeNet Car, Ahmed Body (Fengbo) |
|---|---|---|---|---|---|---|---|
| $N$ | 7225 | 976 | 3131 | 11271 | 16641 | 4096 | $80^3$ |
| $S$ | 64 | 64 | 64 | 64 | 64 | 32 | - |
| $C$ | 128 | 128 | 128 | 128 | 128 | 256 | - |
| Complexity | $\mathcal{O}(10^8)$ | $\mathcal{O}(10^7)$ | $\mathcal{O}(0^8)$ | $\mathcal{O}(10^8)$ | $\mathcal{O}(10^8)$ | $\mathcal{O}(10^8)$ | $\mathcal{O}(10^6)$ |

