# OpenReview forum: "Fengbo: a Clifford Neural Operator pipeline for 3D PDEs in Computational Fluid Dynamics"
_ICLR.cc/2025/Conference — ICLR 2025 Poster_

### Official Review · Reviewer_FKFg · 2024-10-29

**Soundness:** 3
**Presentation:** 3
**Contribution:** 2
**Rating:** 6
**Confidence:** 3

**Summary:**

This submission targets the learning of 3D flow fields together with pressure distributions by using Clifford algebra. This approach has been proposed in previous work, and the submission at hand extends its implementation, and add geometry and physics blocks that seem to primarily aim for up- and down-sampling.

As the paper is largely extending previous work, it does not include and evaluate simpler cases, but directly focuses on 3D flows. Results are shown for flows around obstacle geometries from ShapeNet cars, and the Ahmet body.

The results are mixed: in some cases the proposed architecture seems to perform well, but is outperformed by previous work in others. Especially the classic Unet still seems to do a fairly good job, and probably has a much lower computational workload (and simpler implementation).

Overall, I like the direction of the paper: 3D flows are definitely a challenging topic, and important for practical applications. At the same time, the proposed method does not seem overly convincing to me: it is very tailored towards 3D flows, and seems somewhat incremental given the previous work on Clifford based GNNs. With the results in the paper, I would be hesitant to try this approach, and correspondingly, I also find it difficult to really argue for accepting this paper to ICLR. I think with the current, somewhat narrow scope on 3D pressure (+velocity) the submission would be better suited for a more specialized conference or journal.

**Strengths:**

I think the paper has the following strong points:
- it targets non-trivial flow scenarios in three dimensions
- the ShapeNet cars and especially the Ahmet body are interesting use cases
- a nice range of baselines models is compared to
- the underlying theory is complex

**Weaknesses:**

At the same time, the submission has weak spots:
- the approach seems to be specialized to 3D flows, and I don't see how it would naturally extend to other problems
- the gains in terms of accuracy seem to be mild, which is a pity given the complexity of the approach
- the Clifford algebra comes from previous work, and I have to admit that I don't find it intuitive to work with
- the properties of the baselines are not fully clear (e.g., parameter counts are missing)

**Questions:**

What is the bottleneck that caps the model size at 42M parameters? This does not seem overly large for 3D problems.

Minor, but why are the sizes of the other models in table 2 not listed? How many parameters did they have?

(Very minor recommendation, it's a good idea to give intuition with figures like fig. 2 about the Clifford algebra setup, but figure 3, for example, did not add much information.)

---

> ### Author Response · Authors · 2024-11-19
> **Generalisability to Other Dimensions and Advancements Over Past Research in Clifford Algebra Networks**
>
> We wish to thank the reviewer for their constructive feedback and for praising the direction of the paper and its underlying theory. We address their points below:
>
> ---
>
> #### **1. Extension to other Dimensions**
>
> Although the _paper_ is focused on a 3D problem, the _approach_ itself is not. It is quite the opposite, as Clifford Algebra (CA) Networks are renowned for being applicable to problems of any dimension. This was noted by reviewer __E7Y6__ and demonstrated in several papers, e.g. [here](https://arxiv.org/pdf/2408.13619), where the same ResNet was used for both 2D and 3D Maxwell’s Equations.
>
> Fengbo can be easily recast in 2D by moving layers from 3D to 2D and employing $G(2,0,0)$ instead of $G(3,0,0)$. Since 3D flow estimation is more challenging than 2D, we restricted the paper’s scope to the 3D case, as in Li et al., 2024. Applying the same ideas to a 2D problem would be immediate, especially due to CA's inherent flexibility. We have added the section *"Pipeline Generalisability"* in Appendix C and Tables 5, 6 to clarify this.
>
> ---
>
> #### **2. Accuracy & Complexity**
>
> Fengbo offers several benefits:
>
> - **Joint Estimation**
>   We estimate pressure and velocity fields jointly, treating them as coupled and geometrically meaningful quantities. This was highlighted as a promising future direction by GINO's authors in their rebuttal:
>   > “While in this work we only considered the pressure field so Clifford may not be helpful, it will be interesting to explore using Clifford-FNO […] to address complex 3D geometry when modeling more fields in the future”
>   [See here](https://openreview.net/forum?id=86dXbqT5Ua&noteId=MiMUNcGOTL).
>
> The joint estimation implies:
>
> - **Physical Consistency**
>   Pressure and velocity fields are inherently coupled through Eqs. 1–2. Joint estimation captures this coupling and enables physically consistent predictions, with knowledge of one field improving the other.
>
> - **Higher Efficiency**
>   A unified representation for both fields eliminates the need for a more complex model. This improves efficiency while maintaining accuracy, even for coarse discretisation (Appendix D).
>
> - **Low Computational Complexity**
>   Fengbo uses a single network to manage both fields, with complexity depending on the chosen discretisation. This significantly lowers computational costs. We have expanded Table 4 and added Appendix E to stress this.
>
> Its accuracy is consistently *higher or competitive* with other methods, which often fall short in one or more of the areas above.
> We ought to disagree on the level of complexity of the approach. Fengbo is, in fact, *extremely simple* at the network level. More specifically, we wish to emphasise:
>
> - 60% fewer parameters of the SOTA Neural Operator (NO) model.
> - Only 1 NO against 3 in GINO.
> - just convolutions beside the FNO.
> - No learnt embedding of point clouds onto regular grids (Geo-FNO, GNO)
> - No computationally expensive attention layers (see Appendix E).
> - Robust performance despite simple discretisation, unlike Geo-FNO, ONO, OFormer.
> - No latent space, ensuring fully interpretable outputs.
> ---
>
> #### **3. Contribution over past CA Nets**
>
> Fengbo introduces key advancements over other CA Nets:
>
> - Extending the Clifford NO code to support **any 3D problem** beyond Maxwell’s equations, as it was implemented by Brandstetter et al., 2023, which handled only vectors and bivectors.
> - Introducing an approach to map geometrical quantities to physical quantities, surpassing prior works limited to physics-to-physics or geometry-to-geometry mappings (Brandstetter, Ruhe, Pepe).
> - Novel modeling of 3D unstructured meshes CFD data in as multivectors defined over a 3D regular grid.
> - Original Chi-shaped architecture to perform precise operations based on our modeling choices.
> - Solving a 3D fluid dynamics problem, whereas past CA Nets focused on 2D flows.
>
> ---
>
> #### **4. Parameter Specifications**
>
> The parameter counts for other models were not disclosed by their authors. However, GINO has approximately 100M parameters, as confirmed by its authors. Accordingly, we have moved the discussion on parameter count from Table 2 into the main body of the manuscript and expanded the discussion on model size in Appendix D.
>
> ---
>
> #### **5. Memory**
>
> The reviewer correctly noted that 42M parameters is modest, especially for a 3D setting. The bottleneck arises from processing 6D tensors (of size $ B \times C \times M \times M \times M \times D $, where $B$ = batch size, $C$= channels, $M$ = resolution, and $D$ = dimensionality from CA) via 3D convolutions, which are inherently computationally expensive.
>
> ---
>
> We hope to have thoroughly addressed the concerns raised, particularly those regarding Fengbo’s generalisability to other dimensions and its novelty over previous CA Nets. We trust that our clarifications and the additional evidence provided effectively resolve the identified weaknesses, and we sincerely hope they encourage you to reconsider our work more favorably.

---

> > ### Comment · Reviewer_FKFg · 2024-11-24
> > **Rebuttal**
> >
> > I’d like to thank the authors for their extensive updates and additional explanations. They definitely shed more light on the approach, and make me evaluate the method more positively. I think the discussions here and the additional experiments do improve the submission.
> >
> > I should clarify that with “complexity”, I meant setting up the “Clifford operator” itself. Potentially the NS operators used here could be re-used, but it doesn’t seem simple to me to extend this approach to other PDEs (or even transient flows).
> >
> > I also still see inherent limits to the proposed approach: for me, it’s not more of a “white box” than other models that predict p or directly (I don’t find the Clifford representation intuitive), and it’s not a huge step forward over existing methods. Quite a few papers have that property, but I think it’s important these cases to aim for a clear and honest evaluation of the proposed method. That the paper initially e.g. claimed irregular grids, and down-played the limitations on steady state problems is problematic in my opinion.
> >
> > More score was probably overly harsh initially. I have raised my score to a 5, but should mention that I’m still quite on the edge with this paper. I wouldn’t strongly argue for an accept myself, but I wouldn’t protest if others push for it.

---

> ### Author Response · Authors · 2024-11-24
> **On complexity and clarity**
>
> We wish to genuinely thank the reviewer for reading our rebuttal and revised manuscript carefully, and increasing their score from 3 to a 5. We also really appreciate the time taken to engage in constructive discussion.
>
> Despite the reviewer score being final or not, we would still like to comment for the sake of clarity and for full transparency:
>
> ---
>
> ### **1. Complexity**
>
> We now understand better the complexity issue of the model raised by the reviewer. At theoretical level, Clifford Networks are fundamentally different than other approaches. At implementation level, however, there is *virtually no difference* between a Clifford Neural Operator and a Neural Operator, or, more generally, between a Clifford Neural Network and a classical one: **setting up a Clifford NO is not any different than setting up a standard NO.**
>
> The only practical difference, at network level, is the *embedding* of data within an algebra. This translates into one line of code. For example:
>
> ```x = TensorToGeometric(x, grade = 1, algebra = [1, 1, 1])```
>
> Embeds tensor ```x``` as a grade-1 multivector (i.e. a vector) in the $G(3,0,0)$ algebra: this tells the network explicitly to treat ```x``` as a vector in that algebra. Since there is very little practical difference, **extending our approach to other PDEs or to datasets which include transients would be just as hard/easy to do with our Clifford architecture than to do with a classical one**.
>
> The reviewer is not wrong when mentioning that Clifford Algebra might not appear intuitive to work with, and a basic understanding on it is indeed a prerequisite to work with such architectures. With Clifford Algebra we are moving off the beaten track, but we see this as a strength rather than a weakness.
>
> ---
>
> ### **2. White Box Model**
>
> For full clarity: by white box model we mean two things: 1) our approach works *exclusively* in 3D Euclidean space and 2) that **outputs of the layers in our pipeline are interpretable by design**. For example, if we assign to scalar components the meaning of pressure field (as we do in the second half of the architecture) **they cannot not be interpreted as such**. This is because quantities remain consistent within the same algebra.
>
> To the best of our knowledge, no other pipeline that estimates 2 or 3D flows shows such properties. In fact, it is often quite the opposite, since many tend to work in *latent space* instead, such as GINO, GNO or Geo-FNO.
>
> ---
>
> ### **3. Original Claims**
>
> We trust that the original claims the reviewer is referring to are those pointed out by **gPtN**, on the potential overlook of limitations, and by **TsLb**, on the handling irregular point clouds.
>
> We believe the reviewers raised valid points and we addressed them in full transparency of the review process. About the point raised by reviewer  **gPtN**, we did notice some additional limitations that could be highlighted, for example the computational inefficiency of the Clifford Algebra libraries currently available. About the points raised by reviewer **TsLb**, we conceded that claiming the ability to handle irregular point clouds is conceptually different than discretising them and still operating on regular grids with convolutional layers and FNO.
>
> We view these points as subtle albeit important details that, however, do not affect the overall significance and merits of our work. On the contrary, we consider these clarifications—an integral part of a healthy review process—not only necessary but also valuable, as they have helped to further highlight the strengths of our manuscript.
>
> We wish to state this clearly: we have **no reservations** about acknowledging imperfections in our work, taking the reviewers' comments into account and making corrections when needed. We have addressed with complete integrity and transparency any claim that reviewers found overstated and that we agreed with upon rebuttal.

---

> > ### Comment · Reviewer_FKFg · 2024-11-28
> > **Clarifications**
> >
> > Thank you for the additional explanations. I indeed initially misunderstood the property of the proposed architecture to keep working with the Clifford representation throughout the depth. It is interesting to see the model can give good results while each time "mapping back" to the values from the algebra. I'm not sure what conclusions could be drawn from intermediate values inside the layers of a network though, even if they have a physical meaning. Without a loss at these stages, the learning process probably needs the freedom to perturb these in-betweens in unexpected ways.
> >
> > That the authors commit to being open about the limitations is also good to hear. It is important for the general progress in the field that papers are also clear about limitations of the proposed algorithms.
> >
> > I've further raised my score by one step.

---

### Official Review · Reviewer_sKmQ · 2024-11-01

**Soundness:** 2
**Presentation:** 3
**Contribution:** 2
**Rating:** 6
**Confidence:** 3

**Summary:**

The paper introduces Fengbo, a neural operator pipeline that uses Clifford Algebra to solve 3D PDEs in computational fluid dynamics. Fengbo leverages 3D convolutional and Fourier Neural Operator layers within a Clifford Algebra framework to map 3D geometries to physical fields, such as pressure and velocity. It demonstrates competitive accuracy on CFD datasets with fewer parameters and lower computational complexity, while offering interpretability.

**Strengths:**

* The paper introduces a novel approach by embedding the entire architecture in Clifford Algebra, which allows for a unified treatment of geometric and physical data, enhancing model interpretability and preserving geometric relationships.

* This model provides white-box interpretability by representing intermediate outputs as multivectors, which correlate with physical quantities in 3D space.

**Weaknesses:**

* No comparison with the most advanced deep learning-based methods (e.g. transolver, etc.).

* The main results in the paper show better results with fewer parameters, but not the best performance, and it would be better if the performance could be compared with the same parameter Settings.

* The paper was validated on a limited dataset, and it is hoped that it can be validated on more diverse datasets and tasks (e.g., point cloud, structured mesh, regular grid, etc.), which can be referred to transolver's experimental design.

Minor comments:
* L257"... were the range of the summation of l,m,n isspecified by the kernel size and cin,cout are the ..." has some grammar and typo issues in line 127.

* The definitions of metrics in lines 346 to 363 (formulas (11),(12)) are inconsistent (groud truth and estimated results are used in the denominators respectively).

I would consider revising my rating if the author is able to address my questions and effectively improve on the areas I identified as weaknesses.

**Questions:**

Please refer to weaknesses section for questions.

---

> ### Author Response · Authors · 2024-11-19
> **On the comparison with Transformer models, extending the benchmarking and to different data representation**
>
> We wish to thank the reviewer for acknowledging the merits of our pipeline, its geometric consistency, and interpretability. We address the points raised below:
>
> ### Weaknesses:
>
> 1. **Transformers comparison**
>
>    We recognize the importance of contextualizing our work within the broader literature, including fundamental contributions from the transformer literature. We have addressed this as follows:
>
>    - We added three paragraphs in the *Related Work* section for a better contextualization, *two of them on Transformer-based methods*. We highlighted notable architectures, including the Transolver, as well as their merits and limitations in terms of computational complexity, which is generally quadratic or sub-quadratic.
>    - We compared our pipeline to 6 additional models in **Table 2**, including transformer-based models, doubling the reported results **from 6 to 12 models**.
>    - We studied 7 more models in terms of complexity, discretization convergence, and ability to handle irregular geometries in **Table 4**, doubling the number of models **from 7 to 14** and including transformer-based models such as **Galerkin and Transolver**.
>    - We better contextualized the results in *Section 4.3* by comparing Fengbo to other methods, including Geo-FNO (neural operator), ONO, and OFormer (transformer). These are complex architectures that have been shown to fail on large unstructured meshes like those of the ShapeNet Car dataset. Fengbo, however, still performs exceptionally well with coarse discretization and a simple architecture (as opposed to learned mappings from irregular to regular grids, such as Geo-FNO, or computationally intensive approaches such as ONO and OFormer). This remains true even at very coarse discretization levels, as shown in **Appendix A**.
>    - **We added Appendix E**, where we perform a detailed comparison of Fengbo's complexity with the current transformer-based state-of-the-art, the Transolver. This includes:  **Fig. 14**: A comparison of complexity in the 3D case; **Fig. 15** and **Table 9**: A comparison on the 2D case.
>
> These results demonstrate that Fengbo operates at **two orders of magnitude lower complexity**, even when comparing our 3D with Transolver's 2D, while yielding only slightly higher errors. We are optimistic that these additions have increased the quality of the manuscript, strengthened the case for Fengbo's merits, and better contextualized it with respect to current state-of-the-art models such as the Transolver.
>
> ---
>
> 2. **On model's size**:
>
> Since many models do not include their corresponding number of parameters, we moved the discussion on the model size from Table 2 to in-text only. A thorough discussion on model sizes, both in terms of trainable parameters and memory usage (MB), has been added to the ablations in **Appendix D**.
>
> ---
>
> 3. **On different data representations**
>
> We added **Tables 5 and 6** and the subsection **Pipeline Generalizability** in Appendix C - Implementation Details. Since this paper focuses on 3D CFD, we aligned our approach with the representations used by the *only* two known 3D datasets on this problem: ShapeNet Car and Ahmed Body, both of which employ unstructured meshes. For reference:
>    - GINO (Li et al., 2023) also uses these *two* datasets.
>    - Transolver's experimental setup includes eight datasets, of which *only one is 3D* (ShapeNet Car).
>
>    The data representations in Transolver—point clouds, structured meshes, and regular grids—are primarily applied to 2D datasets. However, Fengbo's preprocessing pipeline is specifically tailored to handle 3D data through the following steps:
>    - **(a)** Sampling irregular point clouds from unstructured meshes.
>    - **(b)** Discretizing the points into a regular grid of resolution $M$.
>    - **(c)** Embedding the data into a multivector form.
>
>   Such steps effectively encompass all three data representations above, and we could deal with them through a combination of the steps above:
>    - **Irregular Point clouds**: Steps (b) and (c) convert them into a regular grid and embed them in multivector form.
>    - **Structured meshes**: Steps (a), (b), and (c) sample points from the mesh, discretize them, and embed them.
>    - **Regular grids**: Step (c) directly embeds them into a multivector form (as already demonstrated in the literature).
>
> This is reported in **Table 6**.
>
>   Furthermore, as Fengbo is built on Clifford Algebra, **it inherently generalizes to problems in any dimension** and within any spatial representation. This adaptability ensures that Fengbo is well-suited to a wide variety of data structures and applications.
>
> ---
>
> 4. **Typos**
>
>    We resolved the typos in the text and in Eq. (11)-(12).
>
> ---
>
> We hope that our responses and revisions address the points raised in a satisfactory manner, as we are confident they have strengthened the contributions of our work. We genuinely hope these points inspire the reviewer to view our work more positively.

---

> > ### Comment · Reviewer_sKmQ · 2024-11-26
> >
> > Apologies for my delayed response, and thank you for the comprehensive rebuttal. I appreciate the comparison of methods such as transformer and the detailed analysis of computational complexity. I think this method is valuable in terms of efficiency and lightweight. Based on this, I will raise my score from 5 to 6 to accept this article.

---

> > > ### Author Response · Authors · 2024-11-26
> > >
> > > We are pleased with the reviewer's satisfaction with our rebuttal. We sincerely appreciate the positive reassessment of our work and the reviewer’s support in recommending our article for acceptance to ICLR 2025.

---

### Official Review · Reviewer_E7Y6 · 2024-11-01

**Soundness:** 2
**Presentation:** 3
**Contribution:** 1
**Rating:** 6
**Confidence:** 5

**Summary:**

The paper presents a new deep learning pipeline to predict the velocity and pressure fields in 3D CFD simulations. The algorithm relies on the use of Clifford algebra layers, a mathematical construction which enables the processing of n-dimensional multivector fields. First, the input fields are upsampled and mixed to translate local to global features. Then, the global information is processed with a FNO-like frequency learning algorithm in a regular voxelized domain. Last, the processed information is decoded to the desired outputs: pressure and velocity fields. This pipeline is tested in the ShapeNet Car and Ahmed Body benchmark datasets and compared with other state-of-the-art architectures.

**Strengths:**

* The Clifford algebra architecture is an interesting inductive bias and is generalizable to more complex multidimensional fields.
* The method has less trainable parameters compared to other techniques.
* The last blocks of the algorithm can be interpreted as velocity and pressure fields, so one can have a visual intuition of how the network learns the final  prediction stage.

**Weaknesses:**

* The method is limited for steady-state simulations, and only tested for low density/viscosity fluids.
* The performance of the method is very similar in error compared to existing techniques.
* The voxelization of the space might be very inefficient with more complex geometries.

**Questions:**

* Line 245: Why is the bivector component left blank in the case of the velocity? The surface information is specially important for the velocity field as it determines the boundary layer dynamics, crucial for drag/lift analysis in aerodynamics.
* The method's claim of being a white-box model may be overstated. While the final layers may provide some intuitive information during the decoding stage, the true learning of the underlying physics primarily occurs in the 3D Clifford FNO block, which still operates within a latent space. Furthermore, how practical is the interpretability of the final layers beyond offering intuition about the predictions? Have the authors observed any relevant phenomena, such as different energetic modes of the pressure and velocity field across each decoding stage? Without any analysis of this regard, the interpretability claim remains largely qualitative rather than quantitative and has little practical use.
* Table 2: Why MeshGraphNets are not tested for the ShapeNet car dataset? The original paper predicts both the pressure and the momentum field of the fluid.
* Line 388: "[...] being the only architecture reported able to do so while jointly estimating the scalar pressure field and the 3D velocity field". Any of the reported architecture can be trivially modified to include an additional output (the velocity field), so I don’t see this as a specific advantage of the proposed methodology.

Final comment: The presented paper proposes almost the same methodology as GINO ("Geometry-Informed Neural Operator for Large-Scale 3D PDEs" paper) but changing the GNO layers with Clifford algebra layers. From an accuracy and novelty perspective, the results offer only incremental improvements. While the interpretability of the final layers might provide some vague intuition, the paper fails to extract any substantial insights from an engineering or mathematical perspective. The only real novelty of the paper is the reduction in number of parameters, which in my view is insufficient for the standards of this venue. For these reasons, I'd rate this paper as marginally below the acceptance threshold.

**Details Of Ethics Concerns:**

I have no ethics concerns.

---

> ### Author Response · Authors · 2024-11-19
> **On the contribution and the meaning of joint estimation in Clifford Algebra Networks**
>
> We wish to thank the reviewer for the detailed analysis of our work, and for acknowledging its generalisability to multidimensional fields. We address the points raised below:
>
> ---
>
> ### Weaknesses:
>
> **1**.  We tested our pipeline on the *two* datasets available for 3D CFD simulations, namely ShapeNet Car and Ahmed Body, which are simulations in air at steady state. This is **consistent with the literature on 3D flows** (see Li 2024, Wu 2024).
>
> **2**. That is valid, but we also emphasised the following key advantages:
>    - Coupling between pressure and velocity (**crucial**, more below).
>    - No need for _ad hoc_ modules to handle irregular grids, which might still fail (Geo-FNO).
>    - Low computational complexity
> - Low number of parameters
>
> The accuracy reported is higher than most and comparable to all other methods, which often fall short in one or more of the areas above.
>
> **3**. We showed in Appendix D how even at very coarse discretisation $M = 40$ we still outperform more sophisticated models such as GNO and Geo-FNO.
>
> ---
>
> ### Questions:
>
> **1. Velocity normals**
>
>  ShapeNet Car datasets include *two meshes*, one outlining the surface of the car, over which the pressure field is defined, and one outlining the volume surrounding the car, over which the velocity field is defined. The bivector component is missing in $V$ since *no information about the normals on the mesh of the velocity field* has been provided, as opposed to $P$. We agree that, if it were to be made available, it would likely correspond to an improvement in the performance.
>
> **2. Interpretability**
>
> We agree that the interpretability remains mostly qualitative in our work, as we have not introduced a quantitative analysis of intermediate layers nor operated on them directly. We are aware of authors who exploited the interpretability of Clifford Layers to perform specific operations on them, most notably:
>    - [This paper](https://shorturl.at/7oCNB) on 3D point clouds,
>    - [This paper](https://shorturl.at/HlZgk) on camera poses,
>
> which, however, perform rigid body motion on geometrical quantities. Operating on intermediate pressure and velocity fields, **in a way that respects the physics of the problem**, is more complex, and would require different operations beyond simple neural layers. We leave this aspect for future research, but *we maintain that end-to-end interpretability, albeit so far qualitative, remains both feasible and an important contribution of our work*.
>
> **3. MeshGraph Net**
>
> We have now included MeshGraph Net in **Table 2**, reporting the relative $L_2$ norm on pressure and velocity fields for the ShapeNet Car dataset, as well as in **Table 4**, discussing its computational complexity and properties.
>
>
> **4. Adaptability of other models for velocity estimation:**
>
>    The reviewer is correct in stating that other models can be adapted to estimate velocity, also jointly with pressure. However, **our joint estimation does more than having multiple outputs at once**: we embed pressure and velocity (as well as all the other inputs) in a mathematical space that treats them as geometrically meaningful quantities **within the unified framework of Clifford Algebra (CA)**.  This is a fundamental element in our pipeline, since it provides valuable inductive bias that keeps estimations consistent with each other, and the reason behind the robust performances with a simple pipeline.
>
>   Authors of GINO already emphasized the potential of CA in [their rebuttal]([https://openreview.net/forum?id=86dXbqT5Ua&noteId=MiMUNcGOTL) :
>    > “While in this work we only considered the pressure field so Clifford may not be helpful, it will be interesting to explore using Clifford-FNO […] to address complex 3D geometry when modeling more fields in the future.”
>
>    We are the first to do so with an approach that pairs novel modeling choices (unstructured meshes as 3D discrete volumes of multivectors) with an original pipeline in a geometry-to-phyics setting.  Thus, **while most networks could be adapted to handle multiple fields, we believe only CA Networks can do so in a way that explicitly couples the physics to the geometry of the problem and keeps predictions consistent**.
>
> ---
>
> We are aware that the reviewer is confident in their judgment of our work. Nonetheless, we hope that our rebuttal has made a stronger case for the contributions of our pipeline, particularly after clarifying the **implications of joint estimation within the CA framework**. These contributions are not just limited to a reduction in the number of parameters, but also include:
> - physically and geometrically consistent joint estimation of quantities.
> - competitive performances.
> - low computational complexity
> - robustness to discretisation
> - original data modelling and network design
>
> We hope that addressing the points made by the reviewer could lead to an overall more positive outlook on our work.

---

> > ### Comment · Reviewer_E7Y6 · 2024-11-25
> >
> > I thank the authors for the assesment and aclaration of my previous comments. I would like to make some comments about the rebuttal.
> >
> > Weaknesses
> >
> > 1. I appreciate that the examples of the paper are well known benchmarks, and I agree with the authors that they are appropriate and enough for the paper. My comment was refered to the fact that, even if the examples are well suited, it is not a method tested for general "3D PDE's in Computational Fluid Dynamics": it's constrained to a reduced set of particular equations (steady state airflow aerodynamics).
> >
> > 3. I thank the authors to clarify that: I missed the explanation in the final Appendix. I also appreciate the extra added figures and ablation studies in the new version.
> >
> > Questions
> >
> > 2. I agree that there is a potential visual interpretation of the last layers due to the geometrical nature of the Clifford operators. However, I feel that the interpretation of those internal representations is useless from the engineering point of view: it doesn't give any relevant insight of the physics of the problem or useful engineering metrics, just "visual intuition". Thus I would say that visual interpretability is a nice feature of the model, but wouldn't claim it to be a "white-box model" (like for example the full Navier Stokes equations).
> >
> > 4. I thank the authors for this comment. The fact that the paper reports incremental results hides out other relevant improvements, like the ones listed by the authors in the final comment.
> >
> > Most of the points have been correctly addressed in the rebuttal. I still believe that the method, while practical, lacks significant novelty and offers primarily incremental results. However, I appreciate the improvements it provides in terms of joint estimation of velocity and pressure, interpretability, and efficiency. Considering these contributions, I have raised my score to 6, which is marginally above the acceptance threshold. That said, it is important to note that many of the paper's strengths are directly attributed to the strict application of Clifford Algebra layers, rather than novel contributions from the paper itself.

---

> > > ### Author Response · Authors · 2024-11-26
> > >
> > > We sincerely thank the reviewer for recognising the value added to our work through our rebuttal and for kindly increasing the score to **6**. We are satisfied with the positive reassessment of our contribution, which, while indeed building on the early Clifford Algebra Networks proposed in 2022, offers some additional insights that also the reviewer could appreciate.

---

### Official Review · Reviewer_TsLb · 2024-11-02

**Soundness:** 3
**Presentation:** 3
**Contribution:** 2
**Rating:** 6
**Confidence:** 4

**Summary:**

This work is essentially the combination of "Clifford Neural Layers for PDE Modeling":

https://arxiv.org/abs/2209.04934

and FNO:

https://arxiv.org/abs/2010.08895

with the extension to computational fluid dynamics (CFD). 3D test cases are considered in this work, with the goal of prediction of the pressure and velocity fields. The complexity of the algorithm, error analysis, and visual comparison between the ground truth and prediction were conducted in this work.

**Strengths:**

High quality of writing and figures. Details explanations. A successful extension of Clifford Neural Layers for PDE Modeling to the Navier-Stokes equations for molding fluid dynamics.

**Weaknesses:**

--> The novelty is limited. This is simply just another application of the Clifford Neural Layers for PDE Modeling paper.

--> In the literature review, the classes of PointNet and PointNet++ for deep learning of CFD have been missed. I suggest that the authors take a look and search on Google Scholar to find those articles and perhaps discuss them. Note that PointNet is suitable for unstructured grids and much lighter than graph neural networks since there is no connectivity between nodes.

--> I disagree with the claim of this manuscript saying that their proposed method is appropriate for irregular grids compared to graph neural networks or PointNet because they still convert irregular grids to Cartesian grids and this definitely introduced errors no matter how much you "carefully" convert these data.

--> Following my previous comment, I believe the information listed in Table 4 is misleading. FNO can be used for irregular geometries if one uses geometric transfer. See the following paper:

https://www.jmlr.org/papers/v24/23-0064.html

On the other hand, the proposed method is not inherently designed for irregular geometries, similar to CNNs and FNOs.

--> As a minor comment, it is better to write L2 as the $L^2$ norm

**Questions:**

--> In Table 2, for FNO, the test error is lower than the train error, how is this possible?

--> In Eqs. 20 and 21, the loss function is a combination of the relative $L^2$ norm with the absolute $L^1$ norm. Mathematically, it does not seem reasonable. How do you justify that?

--> I had some concerns, listed in Weakness. Please address them. Thanks.

---

> ### Author Response · Authors · 2024-11-19
> **On Fengbo's novelty over Clifford Algebra Networks, extending the benchmarking and handling irregular grids**
>
> We would like to thank the reviewer for praising our writing and the level of detail of the results, as well as for their insightful comments. We address them below:
>
> ---
>
> #### **1. Contributions over Clifford Algebra (CA) Networks**
>
> Fengbo is not merely an application paper of past works on a new problem (i.e. 3D flows estimation), since it presents several important contributions over previous CA Nets. These include:
>
> - **Extension to general 3D**
>   We extended the Clifford Neural Operator (NO) code to support *any* 3D problem, and not just Maxwell’s equations as originally designed. This enables the processing of full-grade multivectors and hence its extendability.
>
> - **Geometry-to-Physics Mapping**
>   Fengbo maps geometrical quantities to physical ones, while prior works were limited to physics-to-physics or geometry-to-geometry mappings (Brandstetter, Ruhe, Pepe).
>
> - **Novel Data Representation**
>   We introduced an original way of modeling large-scale 3D CFD unstructured meshes as multivectors defined over 3D voxels. Unstructured meshes were not considered in past CA nets papers.
>
> - **Novel Structure**
>   Fengbo has an original structure:
>   - Parallel blocks to operate on $N_g$ geometrical quantities.
>   - Sum of contributions and mixing through the FNO.
>   - Splitting into $N_p$ parallel blocks for each physical parameter to estimate.
>
>   This design is based on our modeling choices and has been chosen to perform precise operations (Table 1).
> ---
>
> #### **2. Addressing PointNet Methods**
>
> We agree that we have overlooked some classes of methods. To better contextualize our work, we have added, among several others, also PointNet and PointNet++-based models to our discussion, as suggested by the reviewer. Specifically:
>
> - These methods are addressed in **Section 2 (Related Work)**.
> - Their performances when estimating pressure and velocity are included in **Table 2**.
> - Their complexity, ability to deal with irregular grids, and discretisation convergence properties are discussed in **Table 4**.
>
> ---
>
> #### **3. Handling Irregular Grids**
>
> The reviewer is correct in stating that we convert irregular grids to Cartesian grids, much like CNNs and FNOs. When we claim that Fengbo handles irregular grids in **Table 4**, we mean that our pipeline can learn from irregular point clouds accurately and efficiently through discretisation. However, this approach is conceptually different from models like PointNet or MeshGraph Net, which handle irregular grids by design. We understand the reviewer’s concern and have modified **Table 4** for Fengbo accordingly.
>
> ---
>
> #### **4. Geo-FNO Distinction**
>
> The reviewer is correct that the FNO can handle irregular geometries if geometric transfer is employed. However, that would be a Geo-FNO architecture, which is considered a distinct model. The FNO itself still requires a regular grid due to the FFT. We made this distinction clear in **Table 4**.
>
> Note also that Geo-FNO *learns* the mapping from irregular to regular grids but fails on the ShapeNet Car dataset. Fengbo, on the other hand, performs competitively even with a simple deterministic discretisation. This is possible thanks to the **coupling of geometrical and physical quantities** within the unified CA framework and the consequent inductive bias. This is a fundamental point for the success of our approach (see comment above)
>
> ---
>
> #### **Questions**
>
> **1.**   The discrepancy between training and testing errors, while counterintuitive, is not uncommon. We believe this arises from the effect of *regularisation*, which penalises the model’s performance during training to prevent overfitting. This is a likely issue here, given the training set includes only a few hundred instances.
>
> **2.** We understand the reviewer’s concern about mixing different loss terms, as they involve different "units" of measurement. We included the $L_1$ norm after empirically observing its regularising effect, which helped reduce overfitting on the small training set.
>
> - The *relative $L_2$ norm* is essentially an $L_2$ with normalisation factor. It is particularly sensitive to larger errors because it squares the deviations, emphasizing outliers and ensuring stability.
> - The *absolute $L_1$ norm* is less sensitive to outliers and more robust to noise, as it penalises errors linearly. This property encourages sparsity and smoother solutions.
>
> By combining these two components, the loss function balances sensitivity to large deviations with robustness to smaller-scale variations and noise. This is conceptually similar to elastic net regularisation, which combines $L_1$ and $L_2$ penalties on a model’s parameters.
>
> ---
>
> We trust that our responses have helped in making a stronger case for Fengbo’s contributions, especially regarding its novelty and advancements over existing CA Networks. We appreciate that the reviewer’s comments have helped us better contextualise Fengbo within the existing methods and improve the quality of our work.

---

### Official Review · Reviewer_gPtN · 2024-11-03

**Soundness:** 2
**Presentation:** 2
**Contribution:** 2
**Rating:** 6
**Confidence:** 3

**Summary:**

The paper introduces Fengbo, a novel computational pipeline designed to solve 3D partial differential equations (PDEs) specifically for applications in computational fluid dynamics (CFD). Utilizing Clifford Algebra, Fengbo employs an architecture that consists solely of 3D convolutional and Fourier Neural Operator (FNO) layers, effectively modeling the PDE solution process as a clear mapping from geometric representations to the underlying physics of the problem. Despite its relatively simple architecture with only 42 million trainable parameters, Fengbo demonstrates competitive accuracy, outperforming five out of six models previously proposed in the literature for the same dataset. The architecture achieves this with reduced computational complexity compared to graph-based methods while estimating both pressure and velocity fields. A notable feature of Fengbo is its transparency; the output of each layer can be visualized as objects and physical quantities in 3D space, thereby classifying it as a "whitebox" model. By integrating geometry with physics, Fengbo offers an efficient, interpretable, and physics-aware solution for addressing complex PDEs in CFD applications.

**Strengths:**

The paper presents a novel approach to solving 3D partial differential equations (PDEs) in computational fluid dynamics (CFD) through the use of Clifford Algebra. The use of a pipeline that incorporates only 3D convolutional and Fourier Neural Operator (FNO) layers within a Clifford Algebra framework is a creative combination of mathematical structures and neural network architectures. This uniqueness not only offers a fresh perspective but also has the potential to influence future research in both machine learning and PDE solving. The ability to visualize outputs as physical quantities in 3D space transforms the model into a "whitebox" system. This enhances the interpretability of complex models, which is increasingly important in scientific computing, as it allows for better understanding and trust in the results produced by neural networks.

**Weaknesses:**

1. Lack of Comprehensive Benchmarking
2. Absence of Generalization and Scalability Discussion
3. Potential Overlook of Limitations

**Questions:**

1. Please compare with more recent developed methods
2. Please conducts more experiments on different datasets or PDEs.

---

> ### Author Response · Authors · 2024-11-19
> **Expanding Benchmarking, Generalisability considerations, Scalability experiments and Limitations**
>
> We wish to thank the reviewer for the time they took to read our manuscript, for the useful comments they made, and for defining our approach *fresh* and *creative*. We address the weaknesses below:
>
> ---
>
> #### **1. Benchmarking**
>
> We agree that benchmarking could be more comprehensive. We decided to expand it as follows:
>
> 1. We added three paragraphs in the Related Work section for better contextualisation in the literature, discussing PointNet methods, Transformers and GNOT.
> 2. We compared our pipeline to **6 more models** in Table 2, **doubling** the reported results from 6 to 12 models.
> 3. We added results on the relative $L_2$ norm for velocity, in addition to pressure, for 8 models in Table 2.
> 4. We studied **7 more models** in terms of complexity, discretisation convergence, and the ability to handle irregular geometries in Table 4, **doubling** the number of models from 7 to 14.
> 5. Added references to Tables 2 and 4
> 6. Contextualised the results obtained in Section 4.3 by comparing them to other methods: Geo-FNO, ONO, and OFormer are complex architectures that fail on large unstructured meshes such as ShapeNet Car. Geo-FNO, for example, employs a *learned* mapping from irregular to regular meshes. Nonetheless, Fengbo still performs exceptionally well, even with a coarse, deterministic discretisation and a simple architecture. This robustness holds even under harsh discretisation levels, as demonstrated in Appendix D. We believe this robustness stems from the joint estimation of different physical quantities embedded in the same mathematical space, providing an inductive bias that ensures the predicted pressure and velocity fields remain **coupled and consistent**.
>
> 7. We added Appendix E, in which we compare Fengbo’s complexity with the current transformer-based state-of-the-art, i.e., the Transolver, for 2 and 3D cases. We demonstrate we operate at 2 orders of magnitude lower complexity than the Transolver while yielding only slightly higher error.
>
> ---
>
> #### **2. Generalisability and Scalability**
>
> We agree that we can make a stronger argument about Fengbo’s Generalisability and Scalability. For this reason:
>
> 1. We added a discussion on **Generalisability** in Appendix C, including Tables 5 and 6. Clifford Algebra Networks are well-known for their adaptability to any dimension. As a result, Fengbo could be readily transposed to 2D, as well as **extended to any other PDE** in 2 and 3D space. An example is shown [here](https://arxiv.org/pdf/2408.13619), where the same Clifford ResNet is used for both 2D and 3D Maxwell’s Equations. Projecting Fengbo down to 2D is not only possible but also straightforward, especially thanks to CA.
>
> 2. To demonstrate the **Scalability** of our approach, we extended Appendix D and **increased ablations from 1 to 4**. Beside ablations on grid size $M$, we added:
>   - Impact of hidden channels in the 3D Clifford FNO block for both datasets.
>   - Impact of the number of Clifford FNO blocks for both datasets.
>   - Impact of Fourier modes in the FNO for both datasets.
>   - Effect of these variables on the number of parameters and model size.
>
> Showing how performances improve as the model scales.
>
> ---
>
> #### **3. Limitations**
>
> We agree that some limitations might have been overlooked. Hence, we extended the Limitations subsection. Besides the computational bottleneck that arises from 3D convolutions (which prevents testing larger versions of Fengbo), we also discussed:
>
> 1. The intrinsic limitation of CA Networks, which are not optimised for use with underlying tensor structures.
> 2. The hypothesis that Fengbo may not perform as well on datasets that are not as rich as ShapeNet Car and Ahmed Body, since we believe full-grade multivectors play a crucial role in the success of this approach.
>
> ---
>
> #### **Answers Questions**
>
> - **4. Please compare with more recently developed methods:**
>   Addressed in **Benchmarking**.
>
> - **5. Dataset selection:**
>   As the focus of the paper is 3D PDEs for CFD, we employed the **only two known comprehensive datasets** on the topic: ShapeNet Car and Ahmed Body.
>
>   This dataset choice aligns with prior literature, such as:
>   - [This paper](https://arxiv.org/pdf/2309.00583), which also focuses on 3D PDEs and employs *the same two datasets*.
>   - [This paper](https://arxiv.org/pdf/2402.02366), which takes a broader scope but uses *only one 3D dataset* (ShapeNet Car) out of eight studied.
>
> Although limited to two, we are confident these datasets are sufficiently comprehensive, since:
>   - ShapeNet Car accounts for variability and complexity of shape.
>   - Ahmed Body accounts for multiple CFD simulation parameters.
>
> ---
>
> We hope these additions clarify most, if not all, doubts raised by the reviewer. We are confident that the feedback provided has helped us further substantiate the strengths and contributions of Fengbo, and the advantages that Clifford Algebra brings to 3D flow estimation and, more broadly, to the NO literature.

---

> > ### Comment · Reviewer_gPtN · 2024-11-26
> >
> > thank you for the response.

---

### Author Response · Authors · 2024-11-18

We wish to thank the reviewers for their valuable comments, which helped to substantially improve the quality of our work.

The key __strengths__ highlighted include the innovative use of mathematical structures with neural networks, combining Neural Operators (NO) and Clifford Algebra (CA) Networks (__sKmQ, gPtN__). Reviewers appreciated Fengbo’s interpretability and its "white box” properties (__gPtN, E7Y6, sKmQ__). Its ability to estimate challenging 3D flow fields with a lightweight model and its strong theoretical foundation were also praised (__E7Y6, FKFg__).

Recurring points raised in the __weaknesses__ section included:
1. The model’s generalisability to other domains/dimensions, which only __E7Y6__ praised, and its scalability.
2. The lack of more extensive benchmarking, which only __FKFg__ praised.
3. The lack of clarity about the advantage of jointly estimating pressure and velocity.
4. The level of novelty when compared to previously published CA Networks (__TsLB, FKFg__).

Points 1 and 2 demanded further experiments. Points 3 and 4 required a clearer explanation to better emphasise Fengbo’s contributions. We addressed them as follows:

---

### 1. To demonstrate scalability (__gPtN__), we:
- Increased the number of ablations in Appendix D from 1 to 4.
- Studied the impact of the number of hidden channels, blocks, and Fourier modes of the FNO on the accuracy.
- Reported the impact of such variables on model size and the number of parameters.

Regarding generalisability, we added **“Pipeline Generalisability”** in Appendix C and reiterated how CA inherently lends itself to multidimensional treatment. We discussed Fengbo’s implementation in 2D in Table 5, demonstrating its extendability to 2D and other PDEs (__FkFg__). We added Table 6 to discuss Fengbo’s adaptability to various common data structures (__sKmQ__).

---

### 2. To address the limited benchmarking, we:
- Doubled the number of benchmarks in both Table 2 and Table 4, including all models recommended by __TsLb, E7Y6, sKmQ__.
- Included results on the relative L2 norm *also* on the velocity field in Table 2.
- Extended Section 2 with PointNet, Transformers, and GNOT, with a focus on the comparison of Transformer and NO models.
- Added Appendix E on Fengbo’s complexity and its comparison to the Transolver model in 2D and 3D, justifying our focus on NOs (__SkmQ__).
- Discussed Geo-FNO, ONO, and OFormer in Section 4.3.

In terms of datasets, we reiterated that we employed all the datasets available on 3D CFD, i.e. ShapeNet Car and Ahmed Body, consistent with past literature on 3D flows.

---

### 3. On joint estimation:
Jointly estimating pressure and velocity is more complex than simply adapting the architecture to produce two outputs, as raised by __E7Y6__, and is a key contribution that distinguishes Fengbo from the PDEs solvers literature.

We achieve this by embedding quantities within a unified mathematical structure that treats them as elements of the same space, which the network understands and operates in.  The potential of CA Nets to estimate multiple fields had already been noted [here](https://openreview.net/forum?id=86dXbqT5Ua&noteId=MiMUNcGOTL). We are the first to do so with a novel pipeline and data modeling to map the geometry of the PDE to the physics with end-to-end interpretable outputs.

This unified approach in CA is at the core of Fengbo’s many advantages, which are not limited to its small size (__E7Y6__) but also include:
- Low complexity.
- Robustness to discretisation.
- Interpretability
- No need for latent space or learning the deformation from irregular point clouds to regular grids.

Its accuracy is consistently higher or at least competitive with other methods, which often fall short in one or more of the areas above.

---

### 4. Fengbo’s advancements over past CA Nets include:
- Extending the Clifford NO code to support *any* 3D problems beyond Maxwell’s equations, as originally written by Brandstetter 2023, which handles only vectors and bivectors.
- Introducing an approach to map geometrical to physical quantities, advancing past works limited to physics-to-physics or geometry-to-geometry uses (Brandstetter 2023; Ruhe 2024; Pepe 2024).
- Modeling large-scale 3D CFD meshes as $G(3,0,0)$ multivectors defined over a 3D regular grid, while previous approaches only explored structured grids.
- Representing surface normals as bivector elements through the dual operator.
- Proposing an original pipeline with a precise Chi-structure to perform clear operations (grade mixing, voxel filling, upsampling, geometry-to-physics mapping, etc., see Table 1) based on the above modeling choices.
- Solving a 3D fluid dynamics problem, while past CA Nets only focused on 2D flows.

---

We hope this comment, along with our responses to each reviewer, can provide a more comprehensive perspective on the robustness of Fengbo and corroborate its novelty and contributions.

---

### Author Response · Authors · 2024-11-22
**Rebuttal Revision Now Online**

We have uploaded the revised manuscript, incorporating all the edits and clarifications requested as outlined below.

---

### Meta-Review · Area_Chair_jZCG · 2024-12-20

**Metareview:**

The submitted paper proposes method addressing 3D PDEs in CFD and uses Clifford Algebra with an architecture based on 3D convolutions and Fourier Neural Operators. The reviewers appreciated the fact this paper operated off the beaten track with a fresh perspective on the topic with an underlying complex theory, as well as the interpretability aspects, low footprint.

There were some perceived weaknesses (benchmarking, lukewarm performance, missing generalization, limitations to 3D flows), but the authors' answers were generally appreciated by the reviewers, and a consensus on acceptance emerged. The AC concurs.

**Additional Comments On Reviewer Discussion:**

The reviewers engaged with the authors, and discussed the paper with the AC.

---

### Decision · Program_Chairs · 2025-01-22

Accept (Poster)